# FairCareNLP: An AI-driven patient review analyzer for healthcare

**Sayyed Mohammad Pourya Momtaz Esfahani**[1], Davey Seeman[2], Christoffer Dharma[1], Mohammad Noaeen[1], Shion Guha[3], Zahra Shakeri[1,3,4]*

1 Institute of Health Policy, Management and Evaluation, Dalla Lana School of Public Health, University of Toronto, Toronto, Canada, 2 Department of Computer Science, Cornell University, New York, United States of America, 3 Faculty of Information, University of Toronto, Toronto, Canada, 4 Schwartz Reisman Institute, University of Toronto, Toronto, Canada

* zahra.shakeri@utoronto.ca

## Abstract

### Objective

To develop and evaluate an automatic patient review analyzer that applies advanced Natural Language Processing (NLP) and machine learning methods to improve the efficiency, fairness, and accuracy of healthcare feedback analysis.

### Materials and methods

We designed a multi-component pipeline incorporating sentiment analysis, key theme extraction, clinical Named Entity Recognition (NER), and fairness modules. Bias mitigation was addressed through the integration of three complementary approaches: adversarial debiasing, Hard Debiasing, and Iterative Null-space Projection (INLP). Multiple BERT-based models (DistilBERT, BioBERT, RoBERTa-base, BERT-base-uncased) were trained and evaluated under varying hyperparameters and fairness/adversarial loss configurations. Model performance was assessed using accuracy, F1, recall, precision, AUC, Equalized Odds (EOD), and Word Embedding Association Test (WEAT) metrics.

### Results

Adversarial loss ($\lambda_{adv} > 0$) consistently decreased model performance across accuracy, F1, precision, and recall. In contrast, Hard Debiasing and INLP improved WEAT scores while preserving or enhancing other metrics, with INLP yielding the best overall performance. Specifically, INLP with fairness loss improved EOD by 14%, gender WEAT scores by 15%, and achieved slight gains for ethnicity and socioeconomic WEAT scores. The best model achieved accuracy of 0.856, F1 score of 0.812, recall of 0.798, and precision of 0.829. The key theme analysis module identified 82% of expert-labeled themes, though 21% of patient comments lacked expert labels for valence or related attributes.

provided the original author and source are credited.

**Data availability statement:** All data and code supporting the findings of this study are publicly available. The processed datasets and analysis scripts are available on GitHub at https://github.com/HIVE-UofT/Patient-Review-Analyzer.

**Funding:** This research was supported by the Vector Scholarship in Artificial Intelligence (Vector Institute) to PM and the Institute for Pandemics (IFP) at the University of Toronto to ZSH. We also acknowledge funding from the Natural Sciences and Engineering Research Council of Canada (NSERC) through the Canada Research Chairs Program and Discovery Grant (RGPIN-2025-07037 to ZSH). The funders had no role in study design, data collection and analysis, decision to publish, or preparation of the manuscript.

**Competing interests:** The authors have declared that no competing interests exist.

## Discussion

Our results demonstrate the trade-offs between fairness and performance in bias mitigation strategies. While adversarial debiasing reduced predictive accuracy, INLP and Hard Debiasing improved fairness without significant degradation in task performance. Gender bias was easier to mitigate than multi-class sensitive attributes such as ethnicity and income. This difference indicates a need for fairness techniques designed for multi-class sensitive attributes.

## Conclusion

This work presents an NLP pipeline for patient feedback analysis with multiple debiasing strategies. This pipeline improves the fairness and accuracy of insights from unstructured patient reviews and supports inclusive patient-centered care.

## 1 Introduction

Patient feedback is an invaluable resource for healthcare organizations striving to enhance care quality and patient outcomes. Studies have shown that healthcare facilities that actively gather and respond to patient feedback witness significant improvements in key performance indicators. A study using systematic patient feedback in an inpatient psychiatric facility found lower readmission rates compared to national benchmarks: 6.1% at 30 days, 9.5% at 60 days, and 16.4% at 180 days [1]. Furthermore, the World Health Organization (WHO) emphasizes that prioritizing patient-centered care, which heavily relies on understanding and acting upon patient feedback, leads to better health outcomes and higher levels of patient satisfaction [2]. Despite its importance, the vast amount of unstructured patient feedback, often collected through surveys, social media, and online reviews, presents significant challenges in analysis and actionable insight extraction.

The advent of advanced technologies, particularly in the context of Natural Language Processing (NLP) and Artificial Intelligence (AI), offers promising solutions to these challenges. AI-powered tools are increasingly recognized for their ability to process large datasets with enhanced speed and accuracy, far surpassing traditional manual methods [3]. For instance, AI-driven analysis has been shown to reduce data processing time, allowing healthcare providers to respond more promptly and effectively to patient needs [4,5]. Additionally, NLP techniques enable the extraction of nuanced insights from unstructured text, such as identifying key themes and sentiments within patient reviews, which can be crucial for improving healthcare delivery [6].

However, while these technological advancements present substantial opportunities, they also introduce new challenges, particularly regarding the presence of biases in AI models. Biases, whether based on gender, ethnicity, or socioeconomic status, can significantly skew the results of NLP analyses, which can result in the underrepresentation of certain groups and the perpetuation of existing disparities in

healthcare [7]. A recent study found that AI models with unmitigated biases were 40% more likely to misclassify sentiment from minority groups, which highlights the need for robust bias mitigation strategies [8]. Furthermore, research has shown that predictive models trained predominantly on White American data perform worse when applied to African American populations, which emphasizes the need for diverse training datasets [9]. Many studies do not report crucial demographic information. In a systematic review of 164 machine learning studies using Electronic Health Record (EHR) data, 64% did not report race or ethnicity and 92% omitted any information on socioeconomic status [10]. This lack of demographic reporting extends to text-based diagnostic applications. In a systematic review of 78 studies using clinical text for diagnosis, only 35.9% provided race data, and among those, 57.1% described study populations that were majority White [11]. These biases can lead to misdiagnoses and a lack of generalization in minority populations [12]. Addressing these issues is essential for developing equitable healthcare solutions that can accurately analyze sentiments across diverse patient populations and support more personalized and effective treatments. This requires continuous monitoring of AI models for bias, regular updates with diverse datasets, and collaboration between AI developers and healthcare professionals to ensure the technology serves all patients equally.

Given these considerations, our work is driven by the imperative to develop an AI-powered patient review analyzer that not only enhances the efficiency and accuracy of patient feedback analysis but also ensures that the insights generated are unbiased and representative of all demographic groups. This approach aims to contribute to a more equitable healthcare system where patient feedback truly informs and improves care delivery.

Several studies have explored the application of NLP in the analysis of patient reviews, particularly focusing on sentiment analysis and key theme extraction. For example, Greaves et al. [13] demonstrated the effectiveness of sentiment analysis in monitoring patient satisfaction in real time, using social media data as a primary source. Similarly, Nawab et al. [14] leveraged NLP to extract meaningful insights from patient surveys, which identified specific areas requiring improvement in hospital services. Also, studies such as Yuan et al. [15] have applied sentiment and topic analysis to hospital experience comments using NLP techniques, while Feizollah et al.'s scoping review (2025) identified that among 52 studies applying NLP to unstructured patient feedback, sentiment analysis appeared in 32, topic modelling in 15, and text classification in 7. These studies highlight the potential of NLP in transforming unstructured patient feedback into actionable insights. However, they also reveal significant gaps, particularly in addressing biases that may skew analysis results. Research has also focused on extracting key themes from patient reviews using techniques such as Latent Dirichlet Allocation (LDA). While LDA can uncover recurring themes within patient feedback [16], it has limitations, such as treating documents as unordered sets of words and requiring a pre-specified number of topics. These constraints can obscure bias-related issues and misrepresent minority viewpoints, thereby perpetuating health disparities and compromising patient care.

Bias detection and mitigation have emerged as critical areas of concern in the application of AI in healthcare. Bolukbasi et al. [17] introduced methods for debiasing word embeddings to reduce gender and ethnic biases in NLP models. This seminal work sparked further research into algorithmic fairness in healthcare AI. For instance, Chen et al. [18] demonstrated that clinical notes contain implicit biases that can lead to disparities in machine learning predictions across racial groups.

Adversarial debiasing techniques have also been developed to remove protected attributes from model representations through a process in which a predictive model is trained in opposition to an adversary network that attempts to infer sensitive attributes, thus enhancing fairness in sentiment analysis [19]. In the healthcare domain, Zhang et al. [20] applied adversarial debiasing to mitigate gender and age biases in clinical outcome prediction models. Similarly, Pfohl et al. [21] proposed a multi-task learning approach to balance fairness and predictive performance in EHR-based risk models. More recently, FairEHR-CLP utilizes contrastive learning for fairness-aware clinical predictions across multimodal EHR data [22], and the FAME framework optimizes both performance and equity via fairness-aware multimodal embeddings in EHR predictive tasks [23]. Alongside adversarial debiasing, other advanced strategies such as Hard Debiasing [17,24],

which projects pre-trained embeddings onto a bias-neutral subspace, and Iterative Null-space Projection (INLP) [25], which iteratively removes protected attribute information from learned representations, have emerged as complementary post-processing approaches. Despite these advances, there is still a considerable gap in integrating comprehensive bias mitigation strategies into NLP pipelines for healthcare applications. Existing models often address specific biases (e.g., demographic parity in classification outputs [26]; debiasing word embeddings to reduce gender/occupation associations [17]; clinician-in-the-loop bias auditing focused on race or age groups [27]), remain limited in scope and fall short of providing a holistic solution that ensures equity across all demographic groups. Our study builds on these prior efforts by taking incremental steps toward more comprehensive approaches. Rajkomar et al. [28] highlighted the challenges of fairness in machine learning for healthcare, emphasizing the need for careful consideration of data collection, model development, and evaluation processes. Work by Gichoya et al. [29] revealed persistent racial bias in medical imaging AI, which increases the need for stronger debiasing techniques. Moreover, work by Obermeyer et al. [30] uncovered substantial racial bias in a widely used algorithm for predicting health needs, which shows how unchecked AI bias can affect decisions in the health system.

To address these challenges, researchers have proposed frameworks for comprehensive bias assessment and mitigation in healthcare AI. For example, Char et al. [31] outlined ethical principles for the development and deployment of AI in medicine, emphasizing the importance of transparency, accountability, and fairness. Similarly, Ghassemi et al. [32] provided a roadmap for developing equitable and trustworthy clinical AI systems, advocating for interdisciplinary collaboration and rigorous evaluation of fairness metrics. As the field progresses, there is a growing recognition of the need for intersectional approaches to bias mitigation, as highlighted by Buolamwini and Gebru [33] in their work on gender and skin-type bias in facial analysis algorithms. Translating these insights to healthcare AI remains an active area of research, with promising directions including federated learning for privacy-preserving and fair model training [34] and causal inference techniques for unbiased decision support systems [35].

Moreover, the literature suggests that self-selection bias in online reviews remains a significant challenge. This bias arises when the individuals who choose to leave reviews are not representative of the entire patient population, potentially leading to an overemphasis on extreme opinions [36]. Hu et al. [37] demonstrated that online reviews often follow a J-shaped distribution, with an overrepresentation of very positive and very negative reviews. In the healthcare context, this bias can skew perceptions of care quality and patient satisfaction. This issue is compounded by the fact that many NLP models have not been adequately trained on diverse datasets, which limits their ability to generalize across different patient demographics. Larrazabal et al. [38] reported that AI models in healthcare can perform worse for underrepresented groups, which can worsen health disparities. This risk creates a clear need for training data that are diverse and representative in healthcare AI applications.

Our work addresses these gaps by developing FairCareNLP, a comprehensive NLP pipeline that integrates multiple components to enable a holistic analysis of patient reviews. The primary objective of FairCareNLP is to provide an AI-powered tool capable of performing sentiment analysis, key theme extraction, clinical named entity recognition (NER), and bias detection and mitigation in a unified framework. This approach aligns with recent calls for more integrated and ethically-aware AI systems in healthcare, as advocated by Char et al. [31] and Ghassemi et al. [32]. This tool is designed to leverage fine-tuned BERT models such as DistilBERT, BioBERT, RoBERTa-base, and BERT-base-uncased, which are enhanced with advanced bias mitigation techniques to assess patient sentiment and satisfaction accurately. The use of these state-of-the-art language models builds upon the work of Lee et al. [39] on BioBERT and Liu et al. [40] on RoBERTa-base, adapting these powerful models to the specific challenges of healthcare text analysis.

A key component of our methodology is the application of three advanced bias mitigation techniques—Adversarial Debiasing, Hard Debiasing, and Iterative Null-space Projection (INLP). These represent distinct approaches applied at different stages of the model pipeline: adversarial debiasing is an in-processing method that trains the model to avoid encoding protected attributes through adversarial objectives; both Hard Debiasing and INLP are post-processing

techniques—Hard Debiasing adjusts pre-trained embeddings via projection onto a neutral subspace, while INLP iteratively removes protected attribute information from learned representations through null-space projection—after model training but before inference [17,24,25]. By integrating all three strategies within a single NLP pipeline for patient review analysis, rather than applying them in isolation as prior studies have done, our work provides a more comprehensive and rigorous approach to mitigating bias. Together, these complementary methods enable a more comprehensive mitigation of bias throughout the training and inference pipeline, helping ensure our sentiment analysis remains equitable across patient demographics.

Another significant aspect of our work is the integration of a key theme prediction module, which employs the LLaMA language model to identify recurring themes and topics in patient reviews [41]. This module is particularly important for highlighting common concerns and areas for improvement in healthcare services, thereby providing actionable insights that can directly inform healthcare strategies.

The significance of our work lies in its potential to transform how healthcare organizations utilize patient feedback, making a substantial impact in the field of healthcare analytics. By providing a robust and comprehensive tool for analyzing patient reviews, we aim to significantly enhance the accuracy and fairness of insights generated from this data. Our approach not only improves the efficiency of feedback analysis but also ensures that all patient voices are equitably represented in the outcomes, contributing to the development of more responsive and inclusive healthcare systems. This aligns with the vision outlined by Rajkomar et al. [28] for ensuring fairness in machine learning to advance health equity. By addressing the challenges of bias, representativeness, and comprehensive analysis in patient feedback, our research contributes to the broader goal of leveraging AI to enhance healthcare quality and equity, as emphasized by recent studies such as Gichoya et al. [29] and Obermeyer et al. [30]. Ultimately, our work responds to the growing recognition that equitable and actionable insights are essential for improving patient-centered care and healthcare outcomes in an increasingly diverse and complex healthcare landscape.

## 2 Materials and methods

### 2.1 Dataset

This study uses a dataset from patient surveys administered by the National Research Corporation (NRC), a US-based organization located in Ontario, Canada. The surveys include quantitative metrics and open-ended responses about hospital experiences, and they provide self-reported patient feedback. The Investigative Journalism Bureau (IJB), affiliated with the Dalla Lana School of Public Health at the University of Toronto, acquired the de-identified dataset covering seven years through freedom of information requests. The resulting compilation includes more than 120,000 anonymized patient reviews from 45 hospitals in Ontario. These reviews span 2015–2022. Our dataset consists of free-text patient experience comments associated with hospital encounters (e.g., inpatient vs. emergency) and coarse time information (month/year). We intentionally discard exact visit dates and retain only month/year granularity to reduce linkage risk. Because these narratives may contain sensitive health information, we treated the dataset as sensitive throughout the project even when direct identifiers were not present. Prior to analysis, the text records were de-identified by the data provider via manual review, with direct identifiers removed or replaced by placeholder tokens (e.g., 'XXXX...') in the free-text field. Direct identifiers targeted for removal included (but were not limited to) personal names, clinician/staff names, addresses, email addresses, phone numbers, patient identifiers, and other explicit contact/location details. This de-identification was performed through a dedicated manual review process prior to our receiving the data. We did not attempt to re-identify any individual, and no linkage to external datasets was performed. Ontario hospitals and health networks distributed these surveys, which captured various aspects such as hospital attributes (e.g., name, type, units), patient experience (e.g., patient reviews, visit date, sentiment valence), and key themes affecting patient satisfaction (e.g., respect, transportation, access, coordination). The dataset has been annotated to allow for the possibility of multiple key themes being associated with each patient review. Analyses were completed from January to August 2024.

To address the lack of demographic information in our deidentified dataset, we leveraged multiple authoritative sources to construct a representative profile of the population served by the hospitals in our study. We synthesized data from *Statistics Canada*, hospital stays data from the *Canadian Institute for Health Information*, and health care experience survey data from the *Ontario Ministry of Health* to extract the distributions of males and females, different age groups, ethnicities, and socioeconomic factors including Income/Poverty Level, Employment Status, Access to Transportation, and Educational Attainment for the cities in which hospitals were located. Using these distributions, we generated synthetic demographic attributes (e.g., gender, age group, ethnicity) for the reviews at each hospital by randomly assigning labels to comments in proportion to the real-world distributions. For instance, if a hospital's patient population was 60% male and 40% female, we randomly labelled the hospitals' reviews using these probabilities. We then used a combination of socioeconomic distributions and considerations relative to gender, age, and ethnicity to generate socioeconomic data. For example, we adjusted the distribution for ages 65+ to reflect that they are mostly retired or not in the labour force, and for the 0–14 age group, we considered that they are mostly not in the labour force. Based on age groups or ethnicities, we applied changes to the distributions and generated socioeconomic data, which produced the model training dataset. These generated demographic attributes were not used to train the sentiment, theme extraction, or NER models, and they were not used in the bias mitigation techniques. They were incorporated solely to enable the calculation of fairness metrics across demographic groups, ensuring that the evaluation of model outputs could be balanced against realistic population distributions. Because they were assigned independently of the review text, they did not introduce new patterns into the training data or influence the learned language representations. We recognize that this method does not capture linguistic patterns associated with specific demographic groups. This approach assumes that demographic distribution serves as the primary factor for fairness evaluation in this feasibility study. We also used the same method with modified distributions from other cities and countries to generate a mock demographic and socioeconomic dataset. This auxiliary dataset did not include any new or fabricated review text; instead, it provided a set of demographic and socioeconomic attributes constructed from realistic external distributions. Its sole purpose was to act as a comparison group with a different population distribution, simulating how an online reviewing population might differ from the institutional survey population. This allowed us to test the performance of our post-processing technique for mitigating self-selection bias in the propensity scoring module.

## 2.2  Sentiment analysis module

The primary predictive task in this module is sentiment analysis, i.e., classifying patient reviews by sentiment polarity (positive, negative, or neutral). Because demographic biases in model representations can distort sentiment predictions, this section details the debiasing strategies (INLP, Hard Debiasing, and Adversarial Debiasing) integrated into the sentiment analysis pipeline.

### 2.2.1  Data preprocessing.
The preprocessing pipeline involves data cleaning by removing null values, duplicates, and irrelevant characters from the comments, standardizing visit dates by manually extracting various date formats and converting them to a consistent ISO 8601 format, and correcting sentiment valence and key themes using the TextBlob library in Python to correct sentiment valence strings and key themes for consistent spelling. Large language models (LLMs) were used to map hospital units and types to refined categories, and encoding labels and sensitive features involved label encoding for sentiment labels and one-hot encoding for gender, ethnicity, and income. This normalization step used only hospital unit/type strings and did not require transmitting record-level patient narrative text. We tokenized comments using various BERT base models DistilBERT, BioBERT, ROBERTa-base, BERT-base-uncased and tried to compare their performance, chunking long comments to ensure comments do not exceed the models maximum sequence length of 512 tokens, and padding shorter comments to maintain uniform input size.

### 2.2.2  Tokenization and chunking.
To handle long comments effectively, we implemented a flagging and splitting process. This process is crucial for both model input and evaluation, as it ensures that all reviews, regardless of length,

can be processed efficiently. Long reviews are flagged and split into manageable chunks, while shorter reviews are processed as-is. The algorithm for this process is outlined in Fig 1 below:

**2.2.3 Model architecture.** We used four different BERT base models DistilBERT, BioBERT, RoBERTa-base, BERT-base-uncased for sentiment analysis, comparing their performance. The model architecture was modified to incorporate bias mitigation techniques. The last classifier layer was adjusted to match the number of sentiment labels in our dataset, and an adversarial classifier was added to predict protected attributes (gender, ethnicity) from the models internal representations. A gradient reversal layer was implemented between the main model and the adversarial classifier.

**2.2.4 Bias mitigation techniques.** In our research, we explore and implement three powerful bias mitigation techniques to enhance the fairness of our language model: Iterative Null-space Projection (INLP), Hard Debiasing, and Adversarial Debiasing. Each method approaches the problem of bias from a different angle, providing a comprehensive strategy for mitigating various forms of bias in natural language processing models.

**Iterative Null-space Projection (INLP)** is a debiasing technique designed to systematically remove bias from the learned representations in a language model. The key idea behind INLP is to iteratively identify and project out the subspace where the bias manifests itself. This method ensures that the final embeddings do not encode information about the protected attributes, thus mitigating bias in downstream tasks.

The INLP method begins by identifying the bias direction, which is a vector in the embedding space that captures the bias. Mathematically, given an embedding matrix $\mathbf{E} \in \mathbb{R}^{d \times n}$, where $d$ is the dimensionality of the embeddings and $n$ is the number of words, and a bias direction $\mathbf{v} \in \mathbb{R}^d$, the projection of an embedding $\mathbf{e}_i$ onto this bias direction is computed as:

$$\mathbf{p}_i = \frac{\mathbf{e}_i \cdot \mathbf{v}}{\mathbf{v} \cdot \mathbf{v}}\mathbf{v}. \tag{1}$$

The debiased embedding is then obtained by subtracting this projection from the original embedding:

$$\mathbf{e}_i' = \mathbf{e}_i - \mathbf{p}_i = \mathbf{e}_i - \frac{\mathbf{e}_i \cdot \mathbf{v}}{\mathbf{v} \cdot \mathbf{v}}\mathbf{v}. \tag{2}$$

INLP extends this idea by iteratively projecting the embeddings onto the null space of the identified bias directions. After each iteration, a new bias direction is identified using a classifier, and the embeddings are further refined. The iterative update rule is given by:

$$\mathbf{E}^{(k+1)} = \mathbf{E}^{(k)} - \mathbf{E}^{(k)}\mathbf{V}^{(k)}(\mathbf{V}^{(k)})^T, \tag{3}$$

---

**Algorithm 1** Review Flagging and Chunking

**Require:** $review, max\_length = 512$
**Ensure:** Chunked review with flags
 1: $tokens \leftarrow tokenize(review)$
 2: **if** $len(tokens) > max\_length$ **then**
 3:     $chunks \leftarrow split\_into\_chunks(tokens, max\_length)$
 4:     $flagged\_chunks \leftarrow add\_flags(chunks)$
 5: **else**
 6:     $flagged\_chunks \leftarrow [tokens]$
 7: **end if**
 8: **return** $flagged\_chunks$

---

**Fig 1. Pseudocode describing the review flagging and chunking process used in data preprocessing.** Reviews exceeding the model's maximum token limit (512) are divided into smaller segments, each tagged with positional flags to preserve context for subsequent analysis.

where $\mathbf{V}^{(k)}$ represents the matrix of bias directions identified in the *k*-th iteration. By repeating this process, INLP effectively removes the bias, making it difficult for the model to rely on protected attributes in its predictions [25]. In our implementation, we first initialize a BERT model without debiasing. We then use this model to compute the bias directions by analyzing the embeddings of word pairs that are known to exhibit bias. These directions are computed using Principal Component Analysis (PCA) on the difference vectors between such pairs. The resulting bias directions are then integrated into the BERT model, which is reinitialized with an INLP debiasing layer. This layer systematically removes bias from the embeddings during training.

**Hard Debiasing** is another powerful technique for mitigating biases in word embeddings. This method involves two primary steps: (1) neutralization and (2) equalization. The goal is to ensure that the embeddings of words associated with biased pairs (e.g., 'he' and 'she') are equidistant from a neutral point in the embedding space. The process begins by identifying the bias subspace using PCA on a set of biased word pairs. The top principal component(s) capture the bias direction. Given an embedding $\mathbf{e}_i$ and the bias subspace $\mathbf{B}$, the neutralization step projects the embedding onto the subspace orthogonal to $\mathbf{B}$:

$$\mathbf{e}_i' = \mathbf{e}_i - \mathbf{e}_i \mathbf{B}\mathbf{B}^T.$$

(4)

In the equalization step, pairs of words that should be equidistant (e.g., 'he' and 'she') are adjusted to ensure that their embeddings are equidistant from the mean of the pair. For a word pair $(\mathbf{e}_1, \mathbf{e}_2)$, the mean embedding $\mathbf{e}_{eq}$ is calculated as:

$$\mathbf{e}_{eq} = \frac{\mathbf{e}_1' + \mathbf{e}_2'}{2}.$$

(5)

The equalized embeddings are then given by:

$$\mathbf{e}_1'' = \mathbf{e}_{eq} + \frac{\mathbf{e}_1' - \mathbf{e}_{eq}}{\|\mathbf{e}_1' - \mathbf{e}_{eq}\|}, \quad \mathbf{e}_2'' = \mathbf{e}_{eq} + \frac{\mathbf{e}_2' - \mathbf{e}_{eq}\|}{\|\mathbf{e}_2' - \mathbf{e}_{eq}\|}.$$

(6)

This process ensures that the embeddings do not favor one side of the bias spectrum over the other, thereby mitigating bias in the model's predictions [17]. In our work, we compute the bias directions for gender, ethnicity, and income-related word pairs as described above. These directions are then used in a Hard Debiasing layer integrated into our BERT model. During training, this layer continuously adjusts the embeddings to reduce bias according to the principles of neutralization and equalization, resulting in more balanced representations across different demographic groups.

The integration of INLP and Hard Debiasing into our BERT-based sequence classification model is achieved through the inclusion of specialized debiasing layers. After computing the bias directions, we reinitialize the BERT model with these layers, which operate on the embeddings during both the forward pass and the backpropagation process. By doing so, we ensure that the learned representations are less likely to encode information related to protected attributes, thus enhancing the fairness of the model in real-world applications.

**Adversarial Debiasing** is a novel approach that leverages the power of adversarial learning to remove protected attributes from the model's internal representations. In practice, this method trains two models simultaneously: the primary model performs the main prediction task (e.g., classifying review sentiment), while an adversarial network tries to predict sensitive attributes such as gender or ethnicity from the same internal representations. A gradient reversal layer ensures that as the adversary becomes better at detecting these attributes, the main model is pushed to learn representations where such attributes cannot be easily inferred. The result is that the model focuses on task-relevant features while minimizing dependence on protected demographic information, thereby improving fairness across groups [20,42]. This function combines the primary task loss (sentiment analysis), fairness loss, and adversarial loss as follows:

$$L = L_{task} + \lambda_{fair}L_{fair} - \lambda_{adv}L_{adv} \tag{7}$$

where $L_{task}$ is the cross-entropy loss for sentiment classification, $L_{fair}$ is the fairness loss, $L_{adv}$ is the adversarial loss, and $\lambda_{fair}$ and $\lambda_{adv}$ are hyperparameters controlling the strength of fairness and adversarial components. Our custom loss function computes the loss for each chunk and adjusts it based on the number of words in chunks to handle long comments effectively.

To ensure fairness in our model's predictions across different demographic groups, we incorporate a **fairness loss component** based on the principles of demographic parity and equalized odds. This approach aims to minimize disparities in the model performance among various sensitive features. The fairness loss is calculated by measuring the differences in true positive rates (TPR) and false positive rates (FPR) between groups. For a binary classifier $\hat{Y}$ and a sensitive attribute $A$, demographic parity seeks to achieve:

$$P(\hat{Y} = 1 \mid A = 0) = P(\hat{Y} = 1 \mid A = 1) \tag{8}$$

Similarly, equalized odds aims to equalize both TPR and FPR across groups:

$$P(\hat{Y} = 1 \mid Y = 1, A = 0) = P(\hat{Y} = 1 \mid Y = 1, A = 1)$$
$$P(\hat{Y} = 1 \mid Y = 0, A = 0) = P(\hat{Y} = 1 \mid Y = 0, A = 1) \tag{9}$$

where $Y$ represents the true label. To compute the fairness loss, we first calculate the TPR and FPR for each group $g$:

$$\text{TPR}_g = \frac{\text{True Positives}_g}{\text{Positives}_g}, \quad \text{FPR}_g = \frac{\text{False Positives}_g}{\text{Negatives}_g} \tag{10}$$

We then compute the absolute differences in TPR and FPR between groups:

$$\Delta\text{TPR} = |\text{TPR}_0 - \text{TPR}_1|, \quad \Delta\text{FPR} = |\text{FPR}_0 - \text{FPR}_1| \tag{11}$$

These differences are averaged across all labels $L$:

$$\text{avg\_TPR\_diff} = \frac{1}{L}\sum_{l=1}^{L}\Delta\text{TPR}_l, \quad \text{avg\_FPR\_diff} = \frac{1}{L}\sum_{l=1}^{L}\Delta\text{FPR}_l \tag{12}$$

Finally, the fairness loss is calculated as the average of these differences:

$$\text{Fairness Loss} = \frac{\text{avg\_TPR\_diff} + \text{avg\_FPR\_diff}}{2} \tag{13}$$

By incorporating this fairness loss into our overall loss function, we encourage the model to produce predictions that are more consistent across different demographic groups, thereby enhancing the overall fairness of our sentiment analysis system. The combination of INLP, Hard Debiasing, and Adversarial Debiasing provides a comprehensive approach to mitigating various forms of bias in our language model. Each technique addresses bias from a different perspective, allowing us to create a more robust and fair model for real-world applications.

**2.2.5 Training process.** The training process for our bias-mitigated sentiment analysis model incorporates several advanced techniques to optimize both performance and fairness. Specifically, we use a custom loss function that combines

three elements: (1) the standard classification loss for predicting sentiment labels, (2) a fairness loss that encourages the model to perform equally well across demographic groups, and (3) an adversarial loss that trains the model to make it difficult for a secondary network to infer protected attributes such as gender or ethnicity. This comprehensive approach ensures that the model learns to perform sentiment analysis accurately while mitigating biases. To prevent overfitting— where a model performs well on training data but poorly on unseen data—we employ early stopping by monitoring the validation loss. Training is halted when the validation loss stops improving for a predefined number of epochs, helping to achieve the best generalization performance. Moreover, we apply dropout regularization with a rate of 0.1 in the BERT models. In this method, some of the connections in the neural network are randomly dropped during training, which forces the model to learn more robust patterns rather than memorizing noise from the training set.

For optimization, we use the `AdamW` optimizer, an extension of `Adam` that implements weight decay for additional regularization. The weight decay rate is set to 1e-5, providing a balance between model complexity and generalization ability. During training, we continuously apply projective debiasing to word embeddings. This process involves projecting the embeddings onto a subspace orthogonal to the identified bias directions, ensuring that the model learns fair representations throughout the training process. The model parameters are updated based on the gradients of the combined loss function, which includes the task loss, fairness loss, and adversarial loss components. This multi-faceted approach allows us to simultaneously optimize for performance and fairness.

**2.2.6 Evaluation.** To comprehensively assess the effectiveness of our bias mitigation techniques, we employ a multi-faceted evaluation approach. The Word Embedding Association Test (WEAT) is a commonly used metric in bias evaluation literature; it measures the degree of association between sets of target words and attribute words in embedding space by comparing cosine similarities and using a permutation test to assess effect size and significance [43]. Positive WEAT effect sizes indicate stronger stereotypical associations, while values closer to zero suggest reduced bias. We also evaluate the model's fairness using the equalized odds disparity metric, which measures the difference in true positive and false positive rates across protected groups. This metric helps us quantify the model's consistency in performance across different demographic categories.

For performance evaluation, we use standard metrics such as accuracy, F1 score, precision, and recall. These metrics provide a comprehensive view of the model's ability to correctly classify sentiments across various scenarios. To handle long comments that were split into chunks during processing, we employ a special evaluation procedure. We first reassemble the chunks using the flags added during the chunking process. For each reassembled review, we aggregate the probabilities of all its chunks. The final prediction for the review is computed as the weighted average of class probabilities across all chunks, with weights proportional to chunk lengths. We then calculate the evaluation metrics on the reassembled reviews, treating the aggregated prediction in all chunks of a review as a single prediction for that review. This approach provides an evaluation that represents the model performance on complete reviews, even when the reviews are processed in chunks.

## 2.3 Key theme analysis module

After processing the review and extracting sentiments, we have our key theme analysis module. For key theme analysis, we engineered a prompt and wrapped the reviews with that prompt, using LLaMA3 70B to identify the key themes and the corresponding sections in the reviews for those key themes. All LLaMA inference was performed within our institutional compute environment and record-level patient narrative text was not transmitted to external LLM APIs. We used the following prompt:

```
prompt = ("You are analyzing a patient review to identify key themes or areas dis-
cussed in the text. "

"Key themes are specific topics, concerns, or aspects of the healthcare experience that
the patient "
```

```
"mentions or talks about in their review.\n\n"

"Analyze the following patient review and identify all key themes from this list: "

f"{', '.join(KEY_THEMES)}«/monospace>.\n\n"

"Instructions:\n"

"- Identify themes that represent topics, concerns, or areas explicitly mentioned or
discussed in the review\n"

"- A single review may contain multiple themes\n"

"- Match themes based on the content and context of what the patient is describing\n"

"- If no theme from the list matches the content, use 'unknown'\n"

"- For each identified theme, provide a brief description explaining why this theme ap-
plies\n\n"

f"Patient Review:\n{patient_review}\n\n"

"Respond with a JSON object containing a list of identified themes in the format be-
low:\n"

"{\n"

" \"themes\": [\n"

"  {\n"

"  \"theme\": \"\",\"\n"

"  \"description\": \"\"\n"

"}\n"

"]\n"

"}")
```

## 2.4 NER module

After processing the review and extracting sentiment and key themes, we use the Bio-Epidemiology-NER model to tokenize, lemmatize, and extract entities such as disorders, chemicals, and drugs from patient reviews. The following is an example:

```
Review: "The nurse gave me acetaminophen for my migraine, which helped with the pain
but made me feel nauseous." Extracted Entities: Disorders: migraine, pain, nauseous
Chemicals/Drugs: acetaminophen
```

## 2.5 Propensity scoring module

One bias we anticipate within our online review system is selection bias, where certain groups of reviewers may be more likely to leave hospital reviews than others. A notable example of this bias is what is called the J-shaped curve bias, where strongly positive and strongly negative reviewers are more likely to leave reviews than those with neutral experiences. To address this potential bias, we developed a propensity scoring method to resample online respondents to match the distribution of in-person respondents across variables that could introduce selection bias, such as overall hospital experience. This approach ensures that reviewers who were less likely to respond are better represented in the resampled population, leading to a more balanced presence in our key themes classification model.

The variables considered, each potentially contributing to selection bias, were the following: overall hospital experience (valence), gender, age, ethnicity, hospital unit category, income, employment status, access to transportation, and educational attainment. We assumed that in-person reviews have less selection bias than online reviews, so controlling the variables to match the in-person distribution significantly reduces selection bias.

**2.5.1 Propensity scoring computation.** Propensity scores in our context refer to the probability $P(T = 1 \mid X)$ of a given reviewer belonging to the group of online respondents as opposed to in-person respondents based on the set of variables. Our goal was to balance the distribution of these variables to address selection bias.

In order to test the propensity scoring method, we used the synthetic demographic fields described in the Dataset subsection to construct a mock online cohort that approximated the characteristics of online reviewers. These fields did not include review text but were constructed to simulate how online reviewers might differ demographically from hospital-based respondents. We then trained a logistic regression model to predict the propensity scores $P(T = 1 \mid X)$ from the synthetic online cohort and the survey data. The in-person dataset was drawn from the IJB dataset, which represent institutionally collected feedback. Although the dataset contains real review text, it does not specify the exact submission mode for each individual comment (e.g., paper, phone, or digital survey). We removed rows where the unit data was unknown. The model achieved an accuracy of 0.727 and an F1 score of 0.805, with the largest propensity score being 0.993 and the smallest as 0.084. The relevant dummy classifier achieved an accuracy of 0.647 and an F1 score of 0.786. A neural network with two hidden layers of 64 and 32 neurons achieved an F1 score of 0.804 and an accuracy of 0.728, similar to the logistic regression models performance. We chose to continue with the logistic regression model instead of the neural network for its simplicity and interpretability. Fig 2 displays the difference in propensity scores between online and in-person respondents.

**2.5.2 Propensity scoring strategies.** The three primary propensity scoring techniques are propensity score matching, inverse propensity weighting (IPW), and propensity score stratification. We considered each of these methods and ultimately settled on propensity score stratification.

Propensity score matching involves matching each online reviewer to an in-person reviewer with the most similar propensity score. However, this approach may lead to many online reviews being matched to the same in-person review, potentially resulting in an overemphasis on these in-person reviews and causing biased classification.

Both IPW and propensity score stratification involve weighted resampling from the original population. In IPW, the (pre-normalized) resampling weight $w_i$ for a respondent $i$ is defined as follows:

$$w_i = \begin{cases} \frac{1}{e(X_i)} & \text{if } i \text{ is an online respondent,} \\ \frac{1}{1-e(X_i)} & \text{if } i \text{ is an in-person respondent,} \end{cases} \tag{14}$$

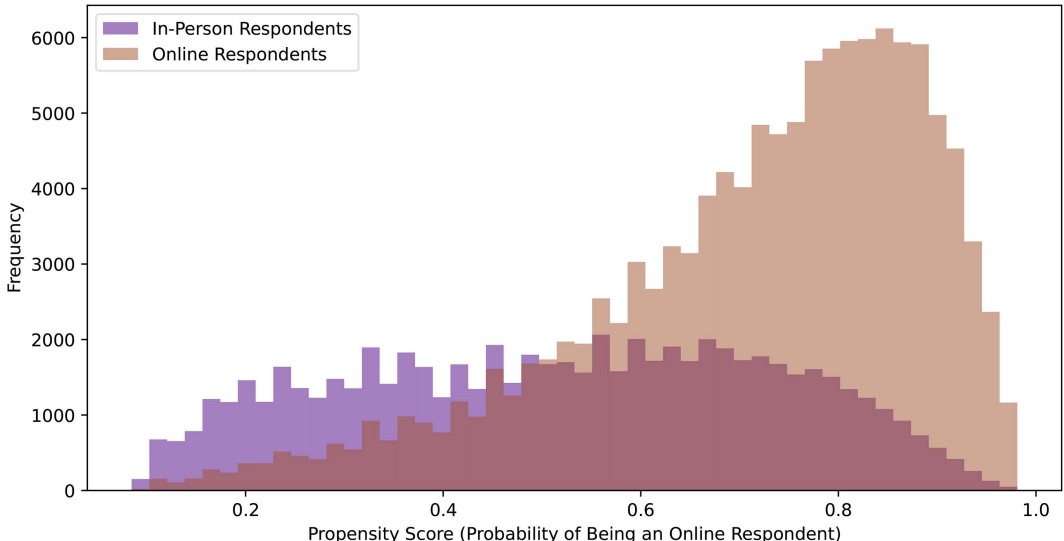

**Fig 2. Propensity Score Distributions: Online vs. In-Person Respondents.** This figure illustrates the propensity score distributions of online and in-person survey respondents, providing insights into the likelihood of participants choosing each survey method based on their characteristics.

where $e(X_i)$ represents the propensity score of respondent $i$. This weighting ensures that after resampling, the groups resemble each other in terms of propensity scores and underlying variable distributions.

Propensity score stratification, on the other hand, involves resampling only from the treatment group (online distribution) and computing weights based on the relative representation of in-person to online respondents across propensity scores. The dataset is split into several strata (10 in our case) based on similar propensity scores. The relative representation $r_s$ of in-person to online respondents within each stratum $s$ is given by:

$$r_s = \frac{\frac{n_{p,s}}{t_p}}{\frac{n_{o,s}}{t_o}},$$

(15)

where $n_{p,s}$ is the number of in-person reviewers within stratum $s$, $t_p$ is the total number of in-person reviewers, $n_{o,s}$ is the number of online reviewers within $s$, and $t_o$ is the total number of online reviewers. The resampling weights $w_i$ are then the normalized relative representations $r_s$ for each respondent's stratum.

Equation 14 defines the weights used in IPW, which ensures balanced representation between online and in-person respondents. Equation 15 is crucial in propensity score stratification, where it calculates the relative representation needed to adjust for the differences in the distribution of propensity scores between the two groups. This stratified approach allows for more nuanced control over the balance between treatment and control groups, ultimately leading to more accurate and less biased results.

## 2.6 System integration and workflow

Although we developed and evaluated each module independently, the FairCareNLP design outlines a future integrated framework for automated patient feedback analysis. In this conceptual system, the sentiment analysis, key theme extraction, and NER modules process patient reviews. The resulting outputs (sentiment labels, thematic categories, and extracted clinical entities) aggregate within a proposed Intelligent Hospital System. This platform serves as a centralized analytics layer to support continuous healthcare quality monitoring. It stores intermediate outputs from all modules to

generate dashboards for healthcare administrators. Before visualization, the propensity scoring and distribution normalization module adjusts for demographic or self-selection biases. This adjustment ensures that reported insights reflect the overall patient population accurately. While this integrated system remains conceptual in the present study, the design demonstrates the intended interoperability of the modules for potential real-world deployment. Fig 3 illustrates the overall workflow of this system. We designed the pipeline components to operate as parallel processing streams rather than a dependent cascade. This architectural choice ensures that the modules extract features from the raw text independently. We modeled the system reliability to minimize error propagation. In a serial cascade, the total system success probability $P_{serial}$ is the product of individual component probabilities $P_i$:

$$P_{serial} = \prod_{i=1}^{n} P_i$$

Such a design leads to rapid performance degradation if one upstream module fails. To mitigate this risk, we implemented a parallel architecture where each module $M$ functions as an independent operator on the input space $X$. This structure ensures that the conditional error probability between modules remains low:

$$P(Error_{Sentiment}|Error_{NER}) \approx P(Error_{Sentiment})$$

Consequently, a failure in entity recognition does not alter the input features for the sentiment classifier or theme extraction modules. This approach minimizes the conditional error dependence between the tasks.

## 2.7 Experimental setup

We conducted all experiments using a computational infrastructure equipped with an NVIDIA Tesla P100 GPU with 16GB of VRAM. The system provided approximately 13GB of system RAM and utilized a multi-core Intel Xeon CPU. We implemented the models using the PyTorch framework. The fine-tuning process for the separate BERT and BioBERT modules required approximately 4 hours cumulative on this GPU for 10 training epochs. We utilized 4-bit quantization and extensive CPU offloading to deploy the 70B parameter LLaMA model for inference due to significant GPU memory constraints. The average end-to-end inference latency for processing a single patient review through the complete pipeline is approximately 15,000 milliseconds.

## 3 Results

### 3.1 Sentiment analysis module

This section presents the performance of different BERT-based models under various configurations designed to mitigate biases in gender, ethnicity, and income within patient reviews of hospitals. We experimented with debiasing techniques such as Iterative Null-space Projection (INLP) and Hard Debiasing, as well as varying adversarial and fairness loss weights ($\lambda_{adv}$ and $\lambda_{fair}$) across multiple learning rates.

**3.1.1 Impact of adversarial and fairness loss.** Introducing adversarial loss ($\lambda_{adv} > 0$) had a pronounced negative effect on the models performance. Across all model's and learning rates, the use of adversarial loss led to a marked decrease in accuracy, F1, precision, and recall. For example, with $\lambda_{adv} = 0.3$ and a learning rate of $1 \times 10^{-3}$, accuracy dropped as low as 0.32, F1 score to 0.12, precision to 0.08, and recall to 0.25. These results suggest that adversarial debiasing significantly hampers the models ability to generalize, likely due to the overly stringent constraints imposed by the adversarial training.

In contrast, incorporating fairness loss ($\lambda_{fair} > 0$) had a more balanced impact. While the primary performance metrics such as accuracy and F1 score remained relatively stable, fairness loss contributed positively to reducing bias, as

                                

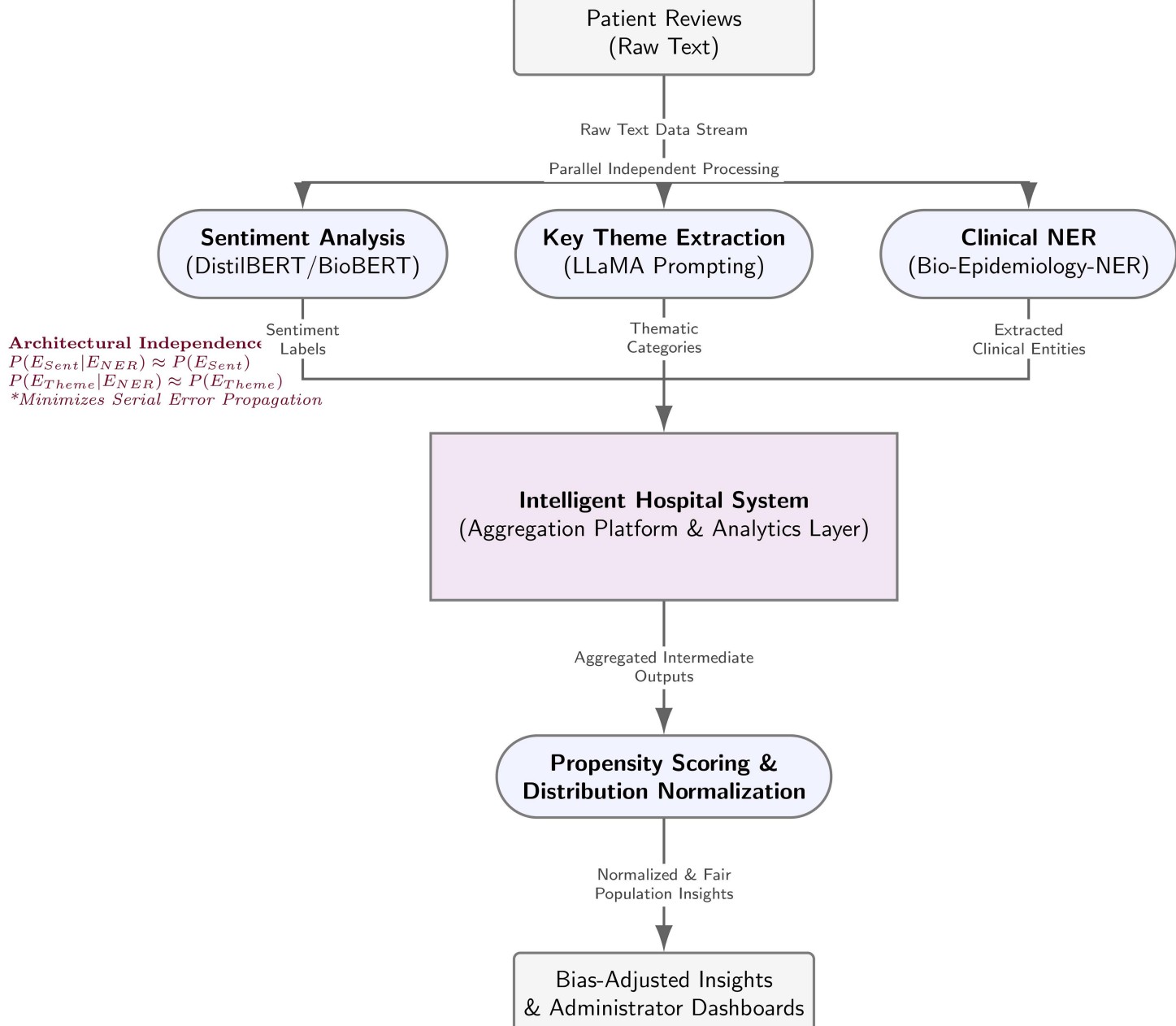

**Fig 3. The high-level modular architecture of the FairCareNLP framework.** The pipeline integrates distinct stages for preprocessing, debiasing, downstream tasks, and evaluation. While debiasing methods are applied upstream, the downstream models for sentiment analysis and key theme extraction operate as parallel, independent processing streams. This design choice ensures that predictive errors within one downstream module do not propagate to parallel tasks.

evidenced by improvements in Equalized Odds Difference (EOD) and Word Embedding Association Test (WEAT) scores. The best results were obtained with $\lambda_{fair} = 0.1$ and $\lambda_{fair} = 0.3$, where EOD improved by up to 14%, and WEAT scores for gender bias improved by up to 15%.

**3.1.2 Comparison of debiasing techniques.** The INLP debiasing method generally produced better bias mitigation results compared to Hard Debiasing. Models incorporating INLP achieved higher EOD improvements and better WEAT scores, especially when applied with fairness loss. For instance, with $\lambda_{fair}$ = 0.3 and INLP, BERT-base-uncased improved EOD by 14% and WEAT by 10%. However, this came with a slight reduction in performance metrics, such as accuracy and F1 score, compared to models without debiasing. On the other hand, Hard Debiasing, while effective, resulted in a slightly lower performance, with accuracy and F1 scores reduced by 2–5% compared to INLP.

**3.1.3 Learning rate effects.** The choice of learning rate played a crucial role in determining model performance. Learning rates larger than $1 \times 10^{-4}$ generally led to poor outcomes, with models failing to improve beyond their initial metrics. At a learning rate of $1 \times 10^{-3}$, models exhibited significantly degraded performance, with accuracy and F1 scores plateauing at low levels. Conversely, a learning rate of $1 \times 10^{-5}$ consistently yielded the best results, with BERT-base-uncased achieving an accuracy of 0.8701 and an F1 score of 0.8682.

**3.1.4 Model comparisons.** Among the various models, BERT-base-uncased exhibited overall better performance, particularly when paired with INLP and fairness loss. For example, BERT-base-uncased with $\lambda_{fair}$ = 0.3 and INLP demonstrated significant improvements in bias mitigation metrics without substantial sacrifices in accuracy or F1 score. In contrast, RoBERTa-base underperformed relative to other models, with metrics generally 3–4% lower, especially when higher learning rates or adversarial loss were applied.

Overall, our results indicate that gender bias was the easiest to reduce, with WEAT and Equalized Odds metrics showing the most consistent improvements across debiasing methods. In contrast, improvements for income and ethnicity were smaller. This is likely due to the fact that gender is a binary attribute, making it easier to equalize, while income and ethnicity are multi-categorical attributes, which complicates fairness adjustments across multiple groups.

The results presented in Table 1 summarize the performance of the most significant model configurations, highlighting the trade-offs between performance metrics and bias mitigation. Additional detailed results for all tested learning rates, debiasing methods, and fairness/adversarial loss configurations are provided in Supporting Information S1 Appendix.

Our experiments show that adversarial loss significantly decreased the models performances. Using fairness metrics with different weights did not substantially affect the overall performance but did help to improve equalized odds. We observed slightly better results with a learning rate of 1e-5, and there was no meaningful difference in performance among the four BERT base models as shown in Fig 4.

Our experiments also reveal distinct trends in WEAT scores across income, gender, and ethnicity during the training process. The trends for each WEAT score type are depicted separately in Fig 5 to highlight the nuances in the behavior of these biases.

The trends observed in Fig 5, demonstrate that income-related WEAT scores exhibit more variability compared to gender and ethnicity WEAT scores. The gender WEAT score shows minor improvements, indicating successful mitigation of

**Table 1. Key Results for BERT Models with Debiasing Techniques.**

| Model | Learning Rate | $\lambda_{fair}$ | $\lambda_{adv}$ | Debiasing Method | Accuracy | F1 | EOD Improvement | WEAT Improvement |
|---|---|---|---|---|---|---|---|---|
| BERT-base-uncased | 1e-5 | 0.0 | 0.0 | None | **0.870** | **0.868** | 0.010 | 0.010 |
| BERT-base-uncased | 1e-5 | 0.3 | 0.0 | INLP | 0.856 | 0.805 | **0.140** | **0.100** |
| BERT-base-uncased | 1e-5 | 0.3 | 0.1 | Hard Debias | 0.850 | 0.797 | 0.100 | 0.060 |
| RoBERTa-base | 1e-5 | 0.3 | 0.0 | INLP | 0.844 | 0.802 | 0.110 | 0.070 |
| RoBERTa-base | 1e-3 | 0.3 | 0.0 | None | 0.330 | 0.125 | 0.010 | 0.010 |
| DistilBERT | 1e-5 | 0.1 | 0.0 | INLP | 0.839 | 0.800 | 0.100 | 0.070 |
| BioBERT | 1e-5 | 0.0 | 0.0 | None | 0.858 | 0.857 | 0.010 | 0.010 |
| BioBERT | 1e-5 | 0.3 | 0.1 | INLP | 0.834 | 0.793 | 0.090 | 0.050 |

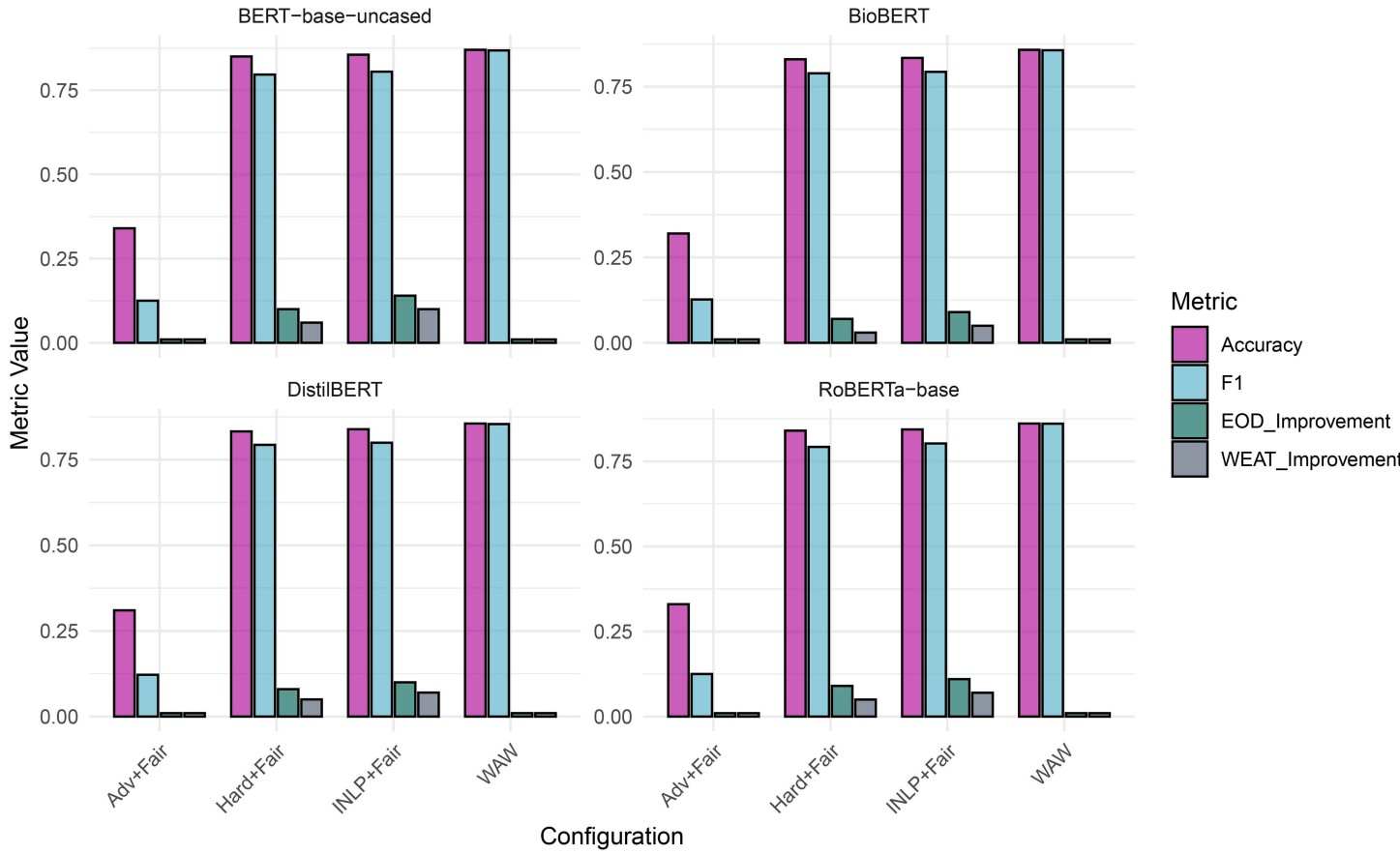

**Fig 4. Comparison of BERT Models with Different Debiasing Techniques.** The figure presents a comprehensive comparison of different BERT models (DistilBERT, BioBERT, RoBERTa-base, BERT-base-uncased) under various configurations, including no debiasing, adversarial debiasing, and fairness with projective debiasing. Metrics such as Accuracy, F1 score, Equalized Odds (EOD), and WEAT are shown. The bar charts illustrate the impact of debiasing techniques on model performance and fairness.

gender bias over time. Ethnicity WEAT scores, while relatively stable, reflect subtle fluctuations, suggesting that the model addresses ethnicity-related bias effectively, though with room for further optimization.

### 3.2 Key theme analysis module

The model identified 82% of the human-labeled key themes, and 21% of the key themes identified by the model were not labeled by experts. Here are two examples of inputs and outputs:

```
Input 1

patient_review="' 1st day in emerge all day asked for water food – received none until
– supper in room Not told why I was put in a small curtained area, no toilet, water.
Trouble breathing – 1 hr before I got oxygen Frightening. "'
```

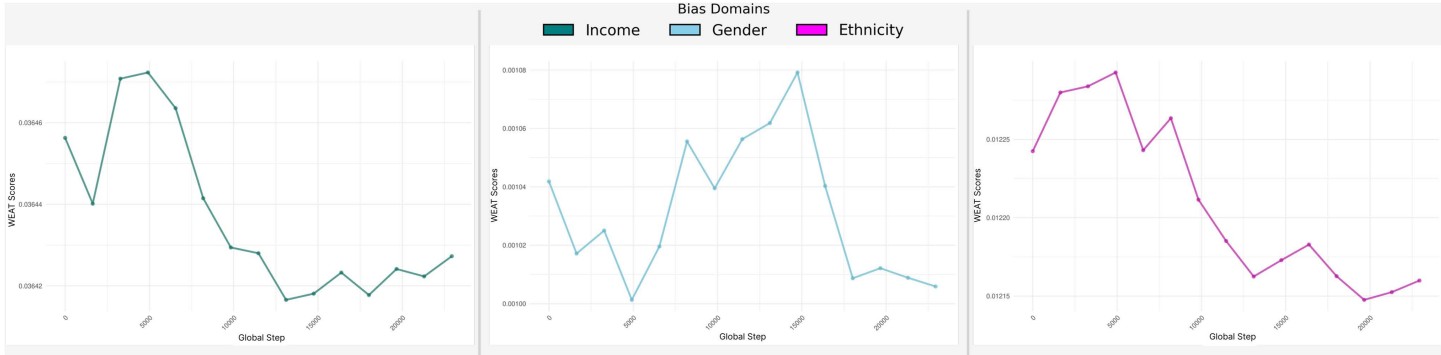

**Fig 5. Trends of Word Embedding Association Test (WEAT) scores across training epochs for different protected domains. Each curve** represents the change in association strength between model representations and sensitive attributes over the course of training. Lower WEAT scores indicate reduced bias in the learned embeddings. The three domains analyzed include Income, Gender, and Ethnicity, capturing socioeconomic, demographic, and racial/ethnic bias dimensions respectively.

Output 1

```
themes: ["theme": "emergency," "description": "Patient experienced trouble breath-
ing and had to wait for oxygen", "theme": "nurse/nurse aide," "description": "Patient
had to ask for basic needs like water and food, but did not receive them promptly",
"theme": "information/education," "description": "Patient was not informed about why
they were put in a small curtained area", "theme": "physical comfort," "description":
"Patient did not receive basic needs like water and food, and had to wait for oxygen
while experiencing trouble breathing"]
```

Input 2

```
patient_review="' 4 ER in 5 weeks 2nd visit should have had IV took 3rd visit 2 week
later? Had swab oil cut 2x 4th visit ask but no response, sign in nurse was rude. Doc-
tor didn't really care XXXXXXX has a good Er I now have several infections on my foot.
1st and 2nd visits same nurse thought I should have had the IV "' Output 2: themes:
["theme": "nurse/nurse aide," "description": "Sign-in nurse was rude and didnt respond
to patient's request, same nurse for 1st and 2nd visits",
```

Output 2

```
themes: ["theme": "emergency," "description": "Multiple visits to the ER, delays
in treatment (IV)", "theme": "continuity/transition," "description": "Same nurse
for 1st and 2nd visits, but didn't provide consistent care", "theme": "infor-
mation/education," "description": "Patient feels doctor didn't really care, and
wasn't provided with proper information or follow-up", "theme": "physical com-
fort," "description": "Patient developed multiple infections on foot due to inade-
quate care"]
```

### 3.3 Named entity recognition (NER) module

We loaded and used the pre-trained `en_ner_bc5cdr_md` model from the `spacy` library. Here is an example of input and output:

**Input**

```
(unreadable) was hurting my ears & I asked for a couple of cotton balls so I could
wrap around tubing (I wasn'tacute; told her name) The nurse replied "Then IÍl have to
tape it on." (I felt as though it was a big chore for her to help me.) As it was I
didn'tacute; need any tape. I worked in health care for over 25 yrs also (I was pre-
scribed codeine & I have an allergy to it.) In addition to the nurse not working to
really help me I also explained & showed the nurse a large rash that was all over my
back (little blisters) but nothing was done about it. It did subside about 1 week lat-
er.
```

**Output**

```
Entity: codeine, Label: CHEMICAL Entity: allergy, Label: DISEASE

Entity: rash, Label: DISEASE
```

### 3.4 Propensity scoring module

To compare IPW with propensity score stratification scoring strategies, we implemented each method and used Cramér's V-statistic to evaluate the association between variable categories and the treatment/control groups in the resampled populations. A lower V-statistic indicates a weaker relationship between the group a participant belongs to and the value of a variable. Before resampling, the average Cramér's V-statistic across all variables was 0.130, with the highest being 0.312 for the 'valence' variable, highlighting the J-shaped curve bias. Both weighting methods successfully reduced Cramér's V-statistic. Propensity score stratification resulted in an average V-statistic of 0.007, with the number for 'valence' dropping to 0.004. Similarly, IPW achieved an average V-statistic of 0.007, with the value for valence also dropping to 0.007. Although both strategies were effective, we plan to use stratification instead of IPW in our future review system. Stratification requires resampling only from online reviews, while IPW also necessitates resampling from in-person reviews. Because we seek to maintain the data integrity of the collected in-person reviews as much as possible, we prefer using stratification. The resulting propensity score distribution after resampling using stratification on the synthetic online data is illustrated in Fig 6. As shown, the propensity score distribution of the resampled online reviews almost exactly matches that of the in-person reviews, except with more overall samples.

When propensity scores become aligned, so do the underlying variables. For example, the mosaic plot in Fig 7 shows how after using propensity score stratification, the employment status distribution for the resampled reviews closely matches that of the in-person reviews. As shown, those not in the labour force were underrepresented in the original online reviews. By using propensity score stratification, people not in the labour force will be reflected with accurate and proportional representation in the key themes classification model.

Finally, propensity scoring can align the overall distributions of variables between the online and in-person data, but it may not create the same alignment within subsets of the online data (e.g., individual hospitals). For example, if one hospital receives many negative reviews, the valence distribution for that hospital is typically more negatively skewed than the

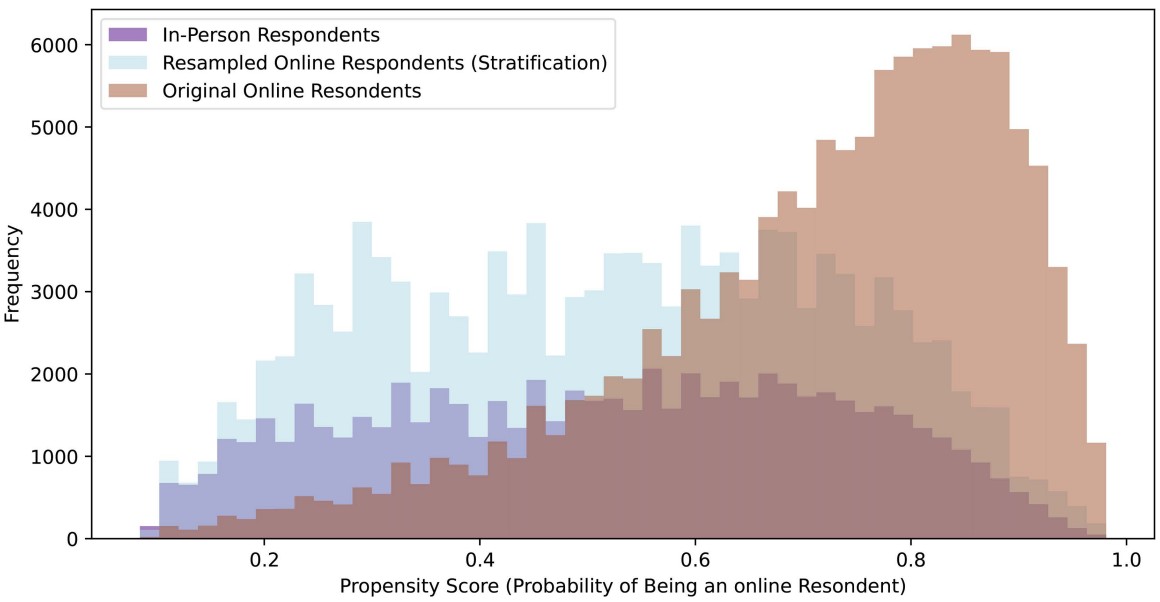

**Fig 6. Propensity Score Distributions: Pre- vs Post-Stratification.** This figure demonstrates the effectiveness of the stratification process in balancing the sample and reducing potential biases in the survey data.

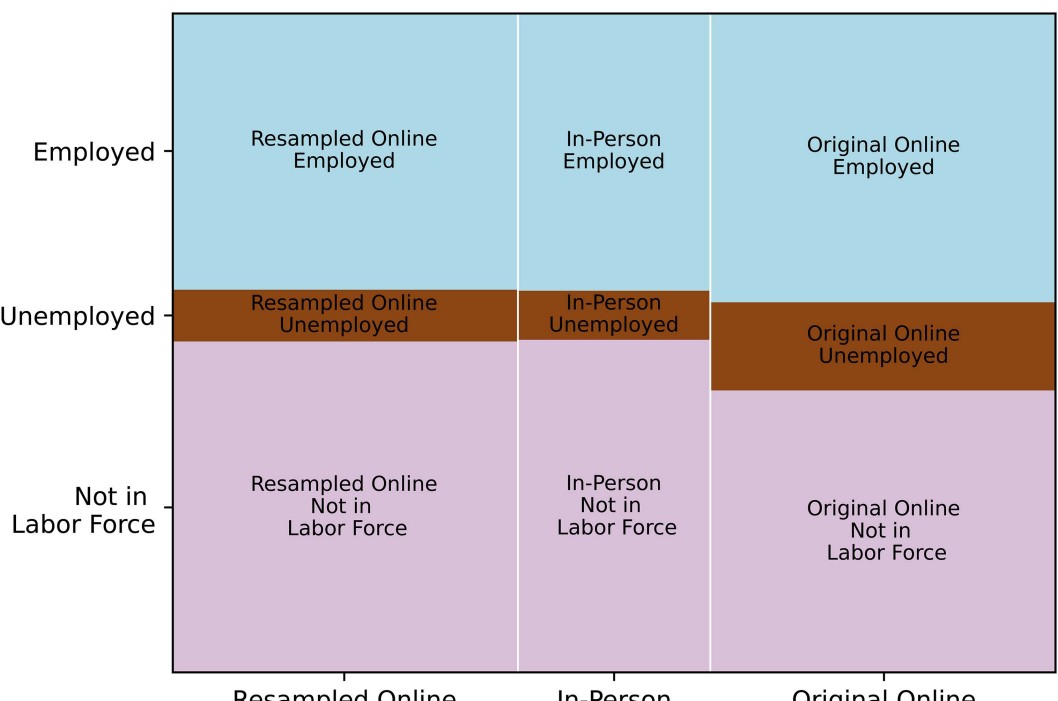

**Fig 7. Employment Status Distribution: Online vs. In-Person Respondents.** This mosaic plot visualizes the relationship between employment status and respondent type, highlighting potential differences in employment patterns between online and in-person survey participants.

valence distribution in the in-person reviews. Thus, our propensity scoring method preserves sub-population trends, such as those observed for specific hospitals, which can be valuable for subsequent sub-population analyses based on the outputs of the key themes classification model on online reviews.

## 4 Discussion

### 4.1 Key findings

A key finding from our experiments was that the choice of debiasing technique had a strong effect on model performance. Adversarial debiasing reduced bias, but it also lowered overall performance on accuracy, F1 score, precision, and recall. This pattern shows a difficult trade-off between bias mitigation and model effectiveness. The challenge is especially important in healthcare applications, where fairness and accuracy are both required [44].

In contrast, the implementation of fairness metrics, even with varying weights, showed that it is possible to improve fairness metrics like Equalized Odds (EOD) without substantially compromising the primary performance metrics. This suggests a promising pathway for enhancing model fairness in healthcare settings, where equitable treatment across different demographic groups is essential [45]. Notably, our results indicated that a learning rate of $1e-5$ consistently delivered optimal results across different BERT-based model's, emphasizing the importance of meticulous hyperparameter tuning in NLP model development [46].

Another important observation was that gender bias proved easier to mitigate compared to income- or ethnicity-related bias. This difference can be explained by the fact that gender, in our study, was treated as a binary attribute, while income and ethnicity were multi-categorical. Debiasing across multiple categories requires balancing performance across several subgroups simultaneously, which increases the complexity of optimization and can dilute improvements. This highlights a limitation of our current approach: while effective for binary attributes, existing debiasing strategies may be less powerful for multi-categorical sensitive features. Future research should therefore explore methods specifically designed for multi-class fairness, such as subgroup-specific constraints, hierarchical debiasing, or causal modeling approaches that can more effectively capture complex population structures.

### 4.2 Challenges

The study faced several challenges that stemmed from the computational demands of the model's and the limits of the available datasets. The BERT-based model's in this analysis required substantial computational resources. This requirement reduced the scope of hyperparameter tuning and limited the number of configurations that the study explored. The constraint highlights the need to balance model complexity with available resources in real-world applications [47].

Another significant challenge was the lack of access to a comprehensive and diverse online patient review dataset. The reliance on synthetic data, while useful for benchmarking, limited the generalizability of our findings. This highlights the critical need for more extensive and diverse datasets in healthcare NLP research to better capture the wide range of patient experiences and improve model robustness [48].

Furthermore, integrating demographic data presented its own set of challenges. Due to privacy concerns and the deidentified nature of our dataset, we had to rely on population estimates and socioeconomic distributions to generate demographic information. While this approach allowed us to address potential biases, it may not fully capture the nuances of individual patient experiences, potentially limiting the model's ability to generalize across different populations [49].

### 4.3 Future directions and improvements

Based on our findings and the challenges encountered, several directions for future research and improvement emerge. A primary need is the acquisition of large, diverse, and representative patient review datasets. Expanding datasets in this way would enable the development of model's that are both robust and generalizable across different demographics, including across provinces and multiple languages. Access to richer data sources would also facilitate benchmarking across studies and help address the limitations associated with relying on synthetic data [50].

Another important avenue concerns the advancement of bias mitigation techniques. Future research should explore approaches that minimize the trade-off between fairness and performance, such as adversarial fairness constraints or domain adaptation strategies. In addition, recent work suggests that incorporating causal reasoning into debiasing frameworks can help identify and correct the specific sources of bias, thereby producing model's that are more equitable as well as more accurate [51].

Methodological refinement through domain-specific adaptation also represents a promising direction. While our study utilized general pre-trained BERT model's, future work should assess the benefits of fine-tuning these model's on healthcare-specific corpora. Tailoring model's to clinical language could improve their understanding of medical terminology and healthcare contexts, leading to more accurate and contextually relevant analyses of patient reviews [52]. Moreover, implementing the analyzer in real-world healthcare settings and conducting longitudinal studies would provide valuable insights into its long-term effects on patient care. Such studies could assess not only whether the system improves patient satisfaction but also its sustained impact on healthcare outcomes over time [53].

Equally critical is the interpretability of these model's. Enhancing transparency in how themes and sentiments are identified will be essential for building trust among healthcare professionals. Techniques such as attention mechanisms, feature importance analysis, and prompt engineering could help reveal how the model's arrive at their conclusions and provide more consistently highly quality outputs, making them more clinically acceptable [54]. These steps would also reduce the possibility of hallucinations or sycophancy from LLMs. Alongside interpretability, expanding the analyzer's capabilities to support multiple languages would significantly extend its applicability, enabling its deployment in diverse healthcare settings and populations. Approaches could include multilingual pre-trained model's or the integration of machine translation techniques [55].

Finally, broader integration, deployment testing, and ethical considerations must be prioritized. Linking patient review analysis with Electronic Health Records (EHRs) could provide a more holistic view of patient experiences by connecting subjective feedback with clinical information [50]. More holistic deployment testing is also required to rule out potential issues such as error propagation. Hospital systems also need to assess the practical utility and actual impact of the workflow on quality of care and patients' experiences. A more thorough evaluation including statistical tests and causal inference would also be beneficial. Lastly, the ethical implications of AI deployment in this domain remain paramount. Future research should emphasize safeguarding patient privacy, ensuring informed consent, and addressing algorithmic bias, all of which are critical for the responsible and equitable use of AI in healthcare [45].

### 4.4 Privacy preservation and synthetic benchmarks

Unstructured patient experience narratives are inherently sensitive. Even after removal of *direct identifiers* (e.g., names) using placeholder tokens such as "XXXX...", there can remain a *residual* risk of re-identification via contextual uniqueness and linkage to outside knowledge (i.e., linkage attacks) [56]. This risk is particularly relevant when free-text narratives are combined with granular metadata (e.g., exact dates, fine geography, or true demographics). For example, consider a single record with fields consistent with our dataset structure:

$$r_i = (h_i, u_i, t_i, \text{month/year}_i),$$

where $h_i$ is a hospital, $u_i$ is a unit (e.g., emergency or inpatient), $t_i$ is free text, and month/year$_i$ is the coarse visit time. A narrative may be de-identified as *"Dr. XXXX explained my results after a rare complication during my pregnancy; my partner could not accompany me due to pandemic restrictions".* Even though direct identifiers are masked, the combination of (i) a rare event, (ii) a specific hospital/unit, and (iii) time period can still be potentially linkable by an adversary with external knowledge. We therefore treat the corpus as sensitive throughout the workflow and do not interpret "XXXX..." replacement as a complete privacy solution on its own.

To address this, we implemented a layered privacy approach designed to materially reduce disclosure risk for unstructured text:

- **Provider de-risking/de-identification.** The dataset was received in de-identified form. The data provider performed manual review using a dedicated process (and personnel) to remove or replace direct identifiers in the narrative text (e.g., personal names, staff names, contact/location details). This process produced placeholders such as "XXXX...".

- **Data minimization.** We intentionally limit metadata granularity. We keep only coarse time fields at the month and year level and do not use exact visit dates in modeling or reporting. We further avoid introducing true individual-level demographics. Instead, we use *synthetic* demographic attributes for fairness benchmarking.

- **Restricted-access processing and screened reporting.** Record-level narratives are processed in a controlled environment with access limited to authorized study personnel, and we do not attempt to re-identify individuals or link to external datasets. To reduce risk in dissemination, the illustrative narrative examples included in this manuscript were reviewed to confirm that no direct identifiers are present.

The protections above focus on de-identification, minimization, governance, and disclosure control. A complementary direction for future deployment is to incorporate *formal* privacy guarantees for *derived releases* (e.g., aggregate statistics, dashboards, or released model checkpoints) using differential privacy (DP) [57]. Given that the dataset we received was already de-identified through multiple rounds of manual review by the data provider focused on direct identifiers, and given that we further reduce linkage risk through data minimization and controlled reporting, this work focuses on methods for modeling and fairness evaluation rather than deployment-time privacy accounting. This includes using month and year only. In future deployment settings where aggregates or model artifacts may be released more broadly, we will incorporate differential privacy mechanisms, including DP noise for released counts and histograms and DP-SGD for training, to provide quantifiable privacy protection for *derived outputs*. A randomized mechanism $M$ is $(\varepsilon, \delta)$-DP if for any neighboring datasets $D \sim D'$ differing by one record and any measurable set of outputs $S$:

$$\Pr[M(D) \in S] \leq e^{\varepsilon} \Pr[M(D') \in S] + \delta. \tag{16}$$

A common deployment-relevant release is a *count* (or histogram) of themes by hospital/unit/year. For example, let

$$f_{h,u,y,\tau}(D) = \sum_{i=1}^{n} \mathbf{1}\{h_i = h, \ u_i = u, \ y_i = y, \ \tau \in T(t_i)\},$$

where $T(t_i)$ maps a narrative to a set of extracted themes and $\tau$ is a theme of interest. Under an event-level neighboring relation, the sensitivity is $\Delta f = 1$, and an $(\varepsilon, 0)$-DP release can be obtained via the Laplace mechanism:

$$\tilde{f}_{h,u,y,\tau}(D) = f_{h,u,y,\tau}(D) + \eta, \qquad \eta \sim \mathrm{Lap}\left(\frac{\Delta f}{\varepsilon}\right) = \mathrm{Lap}\left(\frac{1}{\varepsilon}\right). \tag{17}$$

This provides a quantified bound on the influence of any single narrative on released aggregates. In practice, a deployment would pair DP with standard disclosure controls (e.g., suppressing or aggregating very small strata) to avoid unstable rates and reduce residual linkage risk. Beyond aggregate releases, if the team shares model checkpoints, it can apply DP during training (e.g., DP-SGD). This approach clips per-example gradients and adds calibrated noise. When the pipeline moves from methodological validation to deployment, adding training-time DP with a privacy accountant is an appropriate next step.

Moreover, synthetic demographic attributes in our work support *fairness evaluation* when individual-level demographics are unavailable because of de-identification constraints. These attributes come from external population distributions and are assigned at the hospital level in proportion to those distributions. This design reduces the need to collect, store, or disclose true patient-level demographics, which could increase linkage risk when combined with narrative text. These synthetic demographics do not provide privacy protection for the free-text narratives. They support reproducible fairness benchmarking while following data minimization.

## 4.5 Limitations

This study encountered several limitations primarily due to the complexity and resource-intensive nature of the model's employed. The large-scale BERT-based model's used in our analysis required significant computational resources, which constrained the scope of our experiments. This limitation affected the depth of hyperparameter tuning and the ability to explore a broader range of configurations, such as extended training durations or more fine-grained adjustments to debiasing techniques. We utilized a specific zero-shot prompt for theme extraction to manage computational constraints. A systematic evaluation of output stability requires estimating the variance of the predictive distribution $P(y|x)$. This process necessitates Monte Carlo sampling over $k$ iterations for each review. The total computational cost $C_{total}$ increases linearly with $k$ according to the following relationship:

$$C_{total} \propto k \times T_{inference}(M_{70B})$$

Here, $T_{inference}$ represents the time required for a single pass of the 70-billion parameter LLaMA model. We determined that a sufficiently large $k$ for robust variance estimation was computationally prohibitive for this study. However, the model identified 82% of the themes labeled by human experts. This high concordance rate validates the utility of the current prompt configuration despite the lack of stochastic sensitivity analysis.

Additionally, the lack of access to a comprehensive and diverse online patient review dataset limited our ability to fully evaluate the generalizability of the model's. While synthetic data provided a useful benchmark, real world data that reflects a wide range of patient experiences would likely yield more robust and representative results. Specifically, a primary limitation of this study is our reliance on synthetically generated demographic attributes. Because these labels were assigned based on population distributions rather than ground truth patient data, they have little semantic alignment with the linguistic content of the reviews ($I(X; A) \approx 0$). As a result, the bias detected by the baseline model's reflects spurious algorithmic correlations learned from statistical noise in a finite dataset, rather than systemic societal disparities. While this design limits our ability to measure reductions in dialect based or culturally specific bias, it provides a rigorous mechanistic verification of the pipeline's optimization objective. The results confirm that the system can minimize the mutual information between the model representations ($Z$) and the sensitive attributes ($A$), which acts as a regularizer against overfitting to demographic noise. Future work must validate these components on private, real world clinical datasets where $I(X; A) > 0$, which will allow assessment of the removal of genuine linguistic bias.

Future work must prioritize broader integration and deployment testing. We recommend linking patient review analysis with Electronic Health Records (EHRs) to create a holistic view of patient experiences. This integration connects subjective feedback with clinical information. Researchers must also conduct holistic deployment testing to rule out unforeseen system-level interactions. Hospital administrators should assess the practical utility of the workflow on quality of care.

Transitioning from technical feasibility to real-world deployment requires interdisciplinary collaboration. Future research must involve social scientists and experts in statistical causality. These experts can distinguish between algorithmic artifacts and systemic discrimination. We recommend applying causal inference frameworks to observational clinical data. This application estimates the causal effect of sensitive attributes on model predictions. Such validation ensures that the fairness metrics translate to equitable outcomes for diverse patient populations in varying healthcare settings.

 

# 5 Conclusion

This study has successfully developed an automatic patient review analyzer that leverages advanced Natural Language Processing techniques and machine learning model's to transform unstructured patient feedback into actionable insights. By integrating components such as sentiment analysis, key theme extraction, clinical Named Entity Recognition, and bias mitigation strategies, we have created a comprehensive tool that addresses the complexities of analyzing patient reviews in healthcare settings. This patient analyzer is ideal to be implemented in any systems that wish to incorporate patients' feedback in real time; in lower computational settings, a simpler form of the analyzer (e.g., only the Named Entity Recognition) can be implemented.

The results from the sentiment analysis module demonstrated the complexities of incorporating fairness and debiasing techniques. While adversarial loss ($\lambda_{adv} > 0$) generally led to significant performance degradation, the application of fairness loss ($\lambda_{fair} > 0$) showed promise in improving Equalized Odds Difference (EOD) and reducing bias without a substantial impact on primary performance metrics such as accuracy and F1 score. Among the debiasing methods, Iterative Null-space Projection (INLP) outperformed Hard Debiasing in both bias mitigation and maintaining model performance, particularly when combined with fairness loss. The key theme analysis module, powered by a BERT-based model, was successful in identifying the majority of human-labeled key themes, and it also uncovered additional themes not initially recognized by experts. This highlights the potential of leveraging large language model's (LLMs) for deep insights into patient experiences that may go beyond conventional human analysis. Our NER module, utilizing the `en_ner_bc5cdr_md` model from `spacy`, effectively extracted relevant clinical terms and conditions from patient reviews, providing critical insights into patient-reported symptoms and treatments. The propensity scoring module addressed the challenge of self-selection bias in online reviews. By employing propensity score stratification, we successfully aligned the distribution of online reviews with that of in-person reviews across various variables. This ensured that the analysis of online reviews would be more representative of the broader patient population, thus enhancing the reliability of the key themes classification model.

The implications of this work extend beyond technical advances. It highlights the value of using patient feedback to improve healthcare services and support patient-centered care. The proposed approach provides healthcare organizations with a practical tool for analyzing patient reviews, which can support a more responsive and equitable health system. Future research can expand the dataset to include more diverse patient populations and examine advanced bias mitigation techniques while maintaining model performance. Integration of insights from patient reviews with electronic health records can further improve understanding of patient experiences and outcomes. Our work contributes to the ongoing efforts to improve healthcare delivery by ensuring that patient voices are not only heard but actively shape the quality of care provided. By addressing the challenges of bias and fairness in NLP applications, we can work towards a future where patient feedback is instrumental in driving improvements in healthcare services.

## Supporting information

**S1 Appendix. Detailed performance metrics of sentiment analysis modules.**
(PDF)

## Author contributions

**Conceptualization:** Sayyed Mohammad Pourya Momtaz Esfahani, Davey Seeman, Mohammad Noaeen, Shion Guha, Zahra Shakeri.

**Data curation:** Zahra Shakeri.

**Formal analysis:** Sayyed Mohammad Pourya Momtaz Esfahani, Davey Seeman.

**Funding acquisition:** Shion Guha, Zahra Shakeri.

**Investigation:** Sayyed Mohammad Pourya Momtaz Esfahani, Davey Seeman, Mohammad Noaeen, Zahra Shakeri.

**Methodology:** Sayyed Mohammad Pourya Momtaz Esfahani, Davey Seeman, Christoffer Dharma, Mohammad Noaeen, Shion Guha, Zahra Shakeri.

**Project administration:** Sayyed Mohammad Pourya Momtaz Esfahani, Christoffer Dharma, Mohammad Noaeen, Zahra Shakeri.

**Resources:** Shion Guha, Zahra Shakeri.

**Software:** Sayyed Mohammad Pourya Momtaz Esfahani, Zahra Shakeri.

**Supervision:** Christoffer Dharma, Mohammad Noaeen, Shion Guha, Zahra Shakeri.

**Validation:** Sayyed Mohammad Pourya Momtaz Esfahani, Davey Seeman, Christoffer Dharma, Zahra Shakeri.

**Visualization:** Sayyed Mohammad Pourya Momtaz Esfahani, Zahra Shakeri.

**Writing – original draft:** Sayyed Mohammad Pourya Momtaz Esfahani.

**Writing – review & editing:** Sayyed Mohammad Pourya Momtaz Esfahani, Davey Seeman, Christoffer Dharma, Mohammad Noaeen, Shion Guha, Zahra Shakeri.

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
