## [Decision Letter · Decision Letter 0]

14 Dec 2025

PONE-D-25-60609FairCareNLP: An AI-Driven Patient Review Analyzer for HealthcarePLOS One

Dear Dr. Dharma,

Thank you for submitting your manuscript to PLOS ONE. After careful consideration, we feel that it has merit but does not fully meet PLOS ONE’s publication criteria as it currently stands. Therefore, we invite you to submit a revised version of the manuscript that addresses the points raised during the review process.

Although the manuscript is technically strong, several points should be addressed before it can be considered sufficiently robust for publication. First, the reliance on synthetic demographic attributes constrains ecological validity and reduces confidence in claims about real-world generalizability. Second, the theme extraction stage appears to depend on heuristic LLM prompting. Still, the manuscript does not yet include a structured evaluation of prompt sensitivity, output stability, or hallucination risk, which weakens the interpretability evidence. Third, the current manuscript does not provide adequate end-to-end validation to show how errors may propagate across stages, nor does it demonstrate system robustness under realistic operating conditions. To strengthen the submission, the authors should systematically address the reviewers’ concerns, including, at a minimum, a more transparent and substantive discussion of privacy-preserving design choices and their practical implications.

If applicable, we recommend that you deposit your laboratory protocols in protocols.io to enhance the reproducibility of your results. Protocols.io assigns your protocol its own identifier (DOI) so that it can be cited independently in the future. For instructions see: https://journals.plos.org/plosone/s/submission-guidelines#loc-laboratory-protocols. Additionally, PLOS ONE offers an option for publishing peer-reviewed Lab Protocol articles, which describe protocols hosted on protocols.io. Read more information on sharing protocols at . Additionally, PLOS ONE offers an option for publishing peer-reviewed Lab Protocol articles, which describe protocols hosted on protocols.io. Read more information on sharing protocols at https://plos.org/protocols?utm_medium=editorial-email&utm_source=authorletters&utm_campaign=protocols..

We look forward to receiving your revised manuscript.

Kind regards,

Issa Atoum

Academic Editor

PLOS One

Journal Requirements:

https://journals.plos.org/plosone/s/file?id=ba62/PLOSOne_formatting_sample_title_authors_affiliations.pdexf

2.Thank you for stating in your Funding Statement:

“This research was supported by the Vector Scholarship in Artificial Intelligence (Vector

Institute) to PM and the Institute for Pandemics (IFP) at the University of Toronto to ZSH. We also acknowledge funding from the Natural Sciences and Engineering Research Council of Canada (NSERC) through the Canada Research Chairs Program and Discovery Grant (RGPIN-2025-07037 to ZSH). The funders had no role in study design,

data collection and analysis, decision to publish, or preparation of the manuscript.”

“This research was supported by the Vector Scholarship in Artificial Intelligence (Vector Institute) to PM and the Institute for Pandemics (IFP) at the University of Toronto to ZSH. We also acknowledge funding from the Natural Sciences and Engineering Research Council of Canada (NSERC) through the Canada Research Chairs Program and Discovery Grant (RGPIN-2025-07037 to ZSH). The funders had no role in study design, data collection and analysis, decision to publish, or preparation of the manuscript.”

“This research was supported by the Vector Scholarship in Artificial Intelligence (Vector

Institute) to PM and the Institute for Pandemics (IFP) at the University of Toronto to ZSH. We also acknowledge funding from the Natural Sciences and Engineering Research Council of Canada (NSERC) through the Canada Research Chairs Program and Discovery Grant (RGPIN-2025-07037 to ZSH). The funders had no role in study design,

data collection and analysis, decision to publish, or preparation of the manuscript.”

Additional Editor Comments:

The manuscript should align with PLOS ONE expectations for methods and software by demonstrating practical utility through transparent validation and by ensuring availability, including code, model configurations, and documentation sufficient for independent reproduction. Given that the study is based on deidentified, routinely administered patient experience surveys obtained via freedom-of-information requests, the reporting should follow core STROBE principles for observational data, particularly around sample definition, missingness and labeling limitations, and bounds on inference. If the authors intend to frame the dataset as routinely collected health data in the RECORD sense, they should explicitly justify that choice and map the manuscript to relevant RECORD items without over-claiming compliance. In addition, because demographic attributes are synthetically assigned and used only to compute fairness metrics, the manuscript should clearly delimit what fairness conclusions are supported, and it should avoid causal or real-world equity claims that extend beyond the available evidence.

Reviewers' comments:

Reviewer's Responses to Questions

**Comments to the Author**

1. Is the manuscript technically sound, and do the data support the conclusions?

Reviewer #1: Partly

Reviewer #2: Yes

2. Has the statistical analysis been performed appropriately and rigorously? 

Reviewer #1: N/A

Reviewer #2: Yes

3. Have the authors made all data underlying the findings in their manuscript fully available?

Reviewer #1: Yes

Reviewer #2: Yes

4. Is the manuscript presented in an intelligible fashion and written in standard English?

Reviewer #1: Yes

Reviewer #2: Yes

5. Review Comments to the Author

Reviewer #1: I thank the authors for this good paper that presents FairCareNLP, a comprehensive NLP pipeline for analyzing patient reviews with explicit attention to algorithmic fairness, integrating sentiment analysis, key theme extraction, clinical NER, and bias mitigation strategies within a unified framework. The work is ambitious in scope and well-motivated, particularly given the relevance of bias-aware NLP in healthcare settings. The use of multiple debiasing techniques (INLP, Hard Debiasing, adversarial training) and a broad set of evaluation metrics (including WEAT and Equalized Odds) strengthens the empirical analysis, and the authors are transparent about the observed trade-offs between fairness and predictive performance.

However, several methodological limitations weaken the strength of the conclusions. Most notably, demographic attributes are synthetically generated rather than observed , which raises concerns about the validity and ecological realism of the fairness evaluation; although the authors correctly state that these attributes are used only for assessment, the resulting fairness gains may not transfer to real-world settings with complex, correlated demographic signals. In addition, the key theme extraction relies on a large LLM (LLaMA) in a largely heuristic, prompt-based manner, yet issues such as output stability, hallucination, and sensitivity to prompt design are not systematically examined. Finally, while the pipeline is presented as an integrated system, most components are evaluated independently, leaving open questions about error propagation and end-to-end behavior in deployment scenarios. Overall, the paper represents a solid applied contribution and a useful case study on fairness-aware healthcare NLP, but it would benefit from stronger validation, deeper analysis of LLM reliability, and clearer justification of when such a complex multi-module architecture is preferable to simpler alternatives.

The manuscript would be substantially strengthened by addressing the above points.

Reviewer #2: The paper presents an useful attempt toward achieving a more equitable AI in healthcare by showing that bias mitigation techniques (particularly INLP) can improve fairness without substantially sacrificing accuracy, enabling more inclusive patient-centered care. Existing NLP models contain biases based skewing results and causing underrepresentation of certain patient groups, thus perpetuating existing healthcare disparities and compromises patient care. Authors provide sufficient evidence to substantiate the relevance of this issue. In order to mitigate the problem, they propose a comprehensive NLP pipeline that integrates multiple debiasing techniques (adversarial, Hard Debiasing, INLP) rather than using one in isolation, addresses bias at different stages (in-processing and post-processing), applies to multiple demographic attributes simultaneously, and uses propensity scoring to handle self-selection bias. They test their proposed approach using a dataset of anonymized patient reviews together with synthetic hospital-based demographic attributes.

The validation method demonstrates several notable strengths through its comprehensive and systematic approach. The study employs a large-scale real-world dataset of over 120,000 authentic, anonymized, patient reviews from 45 hospitals, providing substantial empirical grounding. The experimental design is rigorous, testing four different BERT models across multiple hyperparameter configurations (three learning rates, three debiasing techniques, varying loss weights), enabling systematic comparison of fairness-performance trade-offs. Critically, the evaluation framework is multi-dimensional, assessing both traditional performance metrics (accuracy, F1, precision, recall, AUC) and fairness metrics (Equalized Odds Difference, Word Embedding Association Test) across multiple protected attributes (gender, ethnicity, socioeconomic status). The study represents a novel integration of three complementary debiasing strategies applied at different pipeline stages, moving beyond prior work that examined single techniques in isolation. Additionally, the propensity scoring module systematically addresses self-selection bias in online reviews with quantitative validation, and the commitment to reproducibility through public data and code sharing strengthens the contribution.

The most critical weakness undermining the validation is the use of synthetically generated demographic attributes rather than actual patient demographics. These attributes were randomly assigned based on population distributions and (may) have no correlation with the actual review content or writing style, meaning the fairness metrics cannot truly validate whether the models exhibit equitable behavior with real demographic patterns. This fundamental limitation prevents verification of whether bias mitigation actually helps real patients or merely optimizes synthetic benchmarks. The study also lacks external validation—testing only Ontario hospitals with English-language reviews—raising questions about generalizability across regions, healthcare systems, and languages. No real-world deployment testing was conducted with healthcare stakeholders (clinicians or patients) to assess practical utility or actual impact on care quality. The validation suffers from incomplete expert labels (21% missing), limited computational exploration due to resource constraints, absence of statistical significance testing (no confidence intervals or p-values), and no cross-institutional or temporal holdout validation that might reveal overfitting to specific institutional patterns.

In considering whether the draft is ready or not for publication, several considerations need to be taken into account. The validation method is methodologically sound for assessing technical performance but fundamentally limited for validating fairness claims due to reliance on synthetic demographics. It successfully demonstrates that certain debiasing techniques (particularly INLP) can maintain model performance while improving fairness metrics on synthetic benchmarks, establishing technical feasibility. However, the disconnect between synthetic demographic labels and actual patient characteristics means the study cannot confirm whether these techniques would produce equitable outcomes for real diverse patient populations. The work represents an important incremental contribution showing how multiple debiasing strategies can be integrated into healthcare NLP pipelines, but it falls short of validating real-world equity impact. To definitely substantiate fairness claims, the research would need either more explicitly acknowledge the limitations, or alternatively, apply additional tests such as requiring validation on datasets with authentic demographic information (using privacy-preserving methods), cross-institutional testing, multilingual evaluation, and prospective studies involving healthcare stakeholders to assess whether the system genuinely supports more inclusive patient-centered care rather than merely optimizing synthetic fairness metrics.

The reviewer fully acknowledges that conducting the alternative tests would necessarily lead to a better substantiated, but also completely different paper. While the research performed is improvable, asking the authors to engage in the work required to better align with the reviewers' perspective before publication would prevent their unquestionable contribution to immediatly contribute to the discipline, providing a solid and rigorous model for other approaches, and allowing peers to continue this promising line of research. It is necessary to understand that Science is an incremental process made of partial steps. Authors provide a sufficiently solid partial step that others can clearly build upon. Therefore, as long as authors 1) enrich the discussion about privacy preservation, including references to the different approaches (see for example Dwork and colleagues at Harvard on Differential Privacy) and limitations of anonymization (see for example work of Sweeney and associates, also at Harvard), explaining why their chosen approach is the more appropriate one for this paper, and 2) more explicitly state the limitations derived from the use of synthetic demographics, 3) asking for the involvement in future work of social scientists trained in statistical causality (most notably Political Science) as this is the discipline better suited to patch the statistical and causality weaknesses of this paper, then the paper should be considered for publication, and even highlighted in the PONE website.

6. PLOS authors have the option to publish the peer review history of their article (what does this mean?). If published, this will include your full peer review and any attached files.). If published, this will include your full peer review and any attached files.

.

Reviewer #1: No

Reviewer #2: No

---

## [Author Response · Author response to Decision Letter 1]

14 Jan 2026

Please see attached pdf on respond to reviewers for the full response to the reviewer's questions.

---

## [Decision Letter · Decision Letter 1]

3 Feb 2026

PONE-D-25-60609R1FairCareNLP: An AI-Driven Patient Review Analyzer for HealthcarePLOS One

Dear Dr. Dharma,

Thank you for submitting your manuscript to PLOS ONE. After careful consideration, we feel that it has merit but does not fully meet PLOS ONE’s publication criteria as it currently stands. Therefore, we invite you to submit a revised version of the manuscript that addresses the points raised during the review process.

Kindly consider the reviewer's comments on sensitive data and provide detailed responses and actions in the manuscript.==============================

If applicable, we recommend that you deposit your laboratory protocols in protocols.io to enhance the reproducibility of your results. Protocols.io assigns your protocol its own identifier (DOI) so that it can be cited independently in the future. For instructions see: https://journals.plos.org/plosone/s/submission-guidelines#loc-laboratory-protocols. Additionally, PLOS ONE offers an option for publishing peer-reviewed Lab Protocol articles, which describe protocols hosted on protocols.io. Read more information on sharing protocols at . Additionally, PLOS ONE offers an option for publishing peer-reviewed Lab Protocol articles, which describe protocols hosted on protocols.io. Read more information on sharing protocols at https://plos.org/protocols?utm_medium=editorial-email&utm_source=authorletters&utm_campaign=protocols..

We look forward to receiving your revised manuscript.

Kind regards,

Issa Atoum

Academic Editor

PLOS One

Journal Requirements:

Reviewers' comments:

Reviewer's Responses to Questions

**Comments to the Author**

1. If the authors have adequately addressed your comments raised in a previous round of review and you feel that this manuscript is now acceptable for publication, you may indicate that here to bypass the “Comments to the Author” section, enter your conflict of interest statement in the “Confidential to Editor” section, and submit your "Accept" recommendation.

Reviewer #1: All comments have been addressed

Reviewer #2: (No Response)

2. Is the manuscript technically sound, and do the data support the conclusions?

Reviewer #1: Yes

Reviewer #2: Yes

3. Has the statistical analysis been performed appropriately and rigorously? 

Reviewer #1: Yes

Reviewer #2: No

4. Have the authors made all data underlying the findings in their manuscript fully available?

Reviewer #1: Yes

Reviewer #2: No

5. Is the manuscript presented in an intelligible fashion and written in standard English?

Reviewer #1: Yes

Reviewer #2: Yes

6. Review Comments to the Author

Reviewer #1: Thank you for the revised submission. All previously raised concerns have been carefully considered and successfully addressed in the current version of the manuscript.

Reviewer #2: In a prior review, authors have been suggested to appropriately address the sensitive nature of the data. However, in the re-submitted version, this issue remains unaddressed. Provided the sensitive nature of the involved data, and derived high risk, the draft in the current version cannot be considered ready for publication. Authors need to apply effective privacy-preserving techniques capable to ensure that no personal data is leaked. See my prior review with specific details on this issue.

7. PLOS authors have the option to publish the peer review history of their article (what does this mean?). If published, this will include your full peer review and any attached files.). If published, this will include your full peer review and any attached files.

.

Reviewer #1: **Yes:** Benabderrahmane, Sid AhmedBenabderrahmane, Sid Ahmed

Reviewer #2: No

---

## [Author Response · Author response to Decision Letter 2]

19 Feb 2026

We thank the reviewer for clarifying this comment and for the helpful emphasis on privacy protections for unstructured patient narratives. We agree that free-text patient narratives require strong privacy protection and that masking direct identifiers

(e.g., replacing names with ``XXXX\ldots'') does not, by itself, eliminate re-identification risk. In this revision,

we improved the manuscript to (i) explicitly acknowledge residual linkage risk in de-identified narratives,

(ii) clearly describe the layered privacy controls used in our study, and (iii) explain how formal differential

privacy mechanisms would be incorporated in a deployment-oriented next step.

Manuscript changes:

Rewrote Section 4.4 (Privacy Preservation and Synthetic Benchmarks).(Privacy Preservation and Synthetic Benchmarks).

We replaced the prior Section 4.4 with a new discussion including a mock example showing why residualwith a new discussion including a mock example showing why residual

re-identification risk can persist even after ``XXXX\ldots'' masking, and we detail our mitigations, including

provider de-risking via manual review, data minimization (e.g., coarse month/year only and no exact visit dates),

restricted-access processing, and screened reporting examples. We also clarify the role of synthetic demographics

as a fairness benchmarking tool that reduces the need to use true individual-level demographics.

Clarified LLM handling of narrative text (Sections 2.2.1 and 2.3).

Because narrative text is wrapped into the theme-extraction prompt, we explicitly state that LLaMA inference was

performed within our institutional environment and that record-level narrative text was not transmitted to

external LLM APIs. We also clarify that any LLM-assisted unit/type normalization did not involve narrative text.

Revised Data Availability.

We updated Data Availability to remove ambiguity about releasing record-level narratives. The public repository

contains code and non-identifying derived artifacts sufficient to reproduce reported results, while raw narrative

text is not redistributed due to residual re-identification risk inherent to unstructured text.

Formal privacy mechanisms as future deployment work (Section 4.4).).

We added a concise explanation of differential privacy (DP) and an illustrative example (i.e., DP release

of theme counts) to clarify how DP would be applied to derived releases in future deployments. This study does not claim a DP guarantee and suggests DP integration for deployment when releasing aggregates or model checkpoints.

Please find screenshots of the changes in the attached pdf

---

## [Editor Report · Decision Letter 2]

25 Feb 2026

PONE-D-25-60609R2FairCareNLP: An AI-Driven Patient Review Analyzer for HealthcarePLOS One

Dear Dr. Dharma,

Thank you for submitting your manuscript to PLOS ONE. After careful consideration, we feel that it has merit but does not fully meet PLOS ONE’s publication criteria as it currently stands. Therefore, we invite you to submit a revised version of the manuscript that addresses the points raised during the review process.

The revised manuscript could not be sent for review because the comments from the previous round were not fully addressed. As per your request, the submission has been returned to you so that you may correct the issues identified earlier. Once all previous concerns have been fully resolved, the manuscript may be reconsidered for further evaluation.

If applicable, we recommend that you deposit your laboratory protocols in protocols.io to enhance the reproducibility of your results. Protocols.io assigns your protocol its own identifier (DOI) so that it can be cited independently in the future. For instructions see: https://journals.plos.org/plosone/s/submission-guidelines#loc-laboratory-protocols. Additionally, PLOS ONE offers an option for publishing peer-reviewed Lab Protocol articles, which describe protocols hosted on protocols.io. Read more information on sharing protocols at . Additionally, PLOS ONE offers an option for publishing peer-reviewed Lab Protocol articles, which describe protocols hosted on protocols.io. Read more information on sharing protocols at https://plos.org/protocols?utm_medium=editorial-email&utm_source=authorletters&utm_campaign=protocols..

We look forward to receiving your revised manuscript.

Kind regards,

Issa Atoum

Academic Editor

PLOS One
---

## [Author Response · Author response to Decision Letter 3]

27 Feb 2026

Response To Reviewer #2

Reviewer # 2-1: In a prior review, authors have been suggested to appropriately

address the sensitive nature of the data. However, in the re-submitted version, this

issue remains unaddressed. Provided the sensitive nature of the involved data, and

derived high risk, the draft in the current version cannot be considered ready for

publication. Authors need to apply effective privacy-preserving techniques capable

to ensure that no personal data is leaked. See my prior review with specific details

on this issue.

We thank the reviewer for clarifying this comment and for the helpful emphasis on privacy

protections for unstructured patient narratives. We agree that free-text patient narratives re-

quire strong privacy protection and that masking direct identifiers (e.g., replacing names with

“XXXX. . . ”) does not, by itself, eliminate re-identification risk. In this revision, we improved the

manuscript to (i) explicitly acknowledge residual linkage risk in de-identified narratives, (ii) clearly

describe the layered privacy controls used in our study, and (iii) explain how formal differential

privacy mechanisms would be incorporated in a deployment-oriented next step.

Manuscript changes: (1) Rewrote Section 4.4 (Privacy Preservation and Synthetic(Privacy Preservation and Synthetic

Benchmarks). We replaced the prior Section 4.4 with a new discussion including a mock examplewith a new discussion including a mock example

showing why residual re-identification risk can persist even after “XXXX. . . ” masking, and we detail

our mitigations, including provider de-risking via manual review, data minimization (e.g., coarse

month/year only and no exact visit dates), restricted-access processing, and screened reporting

examples. We also clarify the role of synthetic demographics as a fairness benchmarking tool that

reduces the need to use true individual-level demographics.

(2) Clarified LLM handling of narrative text (Sections 2.2.1 and 2.3). Because narra-

tive text is wrapped into the theme-extraction prompt, we explicitly state that LLaMA inference

was performed within our institutional environment and that record-level narrative text was not

transmitted to external LLM APIs. We also clarify that any LLM-assisted unit/type normalization

did not involve narrative text.

(3) Revised Data Availability. We updated Data Availability to remove ambiguity about

releasing record-level narratives. The public repository contains code and non-identifying derived

artifacts sufficient to reproduce reported results, while raw narrative text is not redistributed due

to residual re-identification risk inherent to unstructured text.

(4) Formal privacy mechanisms as future deployment work (). We added(). We added

a concise explanation of differential privacy (DP) and an illustrative example (i.e., DP release of

theme counts) to clarify how DP would be applied to derived releases in future deployments. This

study does not claim a DP guarantee and suggests DP integration for deployment when releasing

aggregates or model checkpoints.

---

## [Decision Letter · Decision Letter 3]

9 Mar 2026

FairCareNLP: An AI-Driven Patient Review Analyzer for Healthcare

PONE-D-25-60609R3

Dear Dr. Dharma,

We’re pleased to inform you that your manuscript has been judged scientifically suitable for publication and will be formally accepted for publication once it meets all outstanding technical requirements.

An invoice will be generated when your article is formally accepted. Please note, if your institution has a publishing partnership with PLOS and your article meets the relevant criteria, all or part of your publication costs will be covered. Please make sure your user information is up-to-date by logging into Editorial Manager at Editorial Manager® and clicking the ‘Update My Information' link at the top of the page. For questions related to billing, please contact  and clicking the ‘Update My Information' link at the top of the page. For questions related to billing, please contact billing support..

Kind regards,

Issa Atoum

Academic Editor

PLOS One

Additional Editor Comments (optional):

Reviewers' comments:

Reviewer's Responses to Questions

**Comments to the Author**

1. If the authors have adequately addressed your comments raised in a previous round of review and you feel that this manuscript is now acceptable for publication, you may indicate that here to bypass the “Comments to the Author” section, enter your conflict of interest statement in the “Confidential to Editor” section, and submit your "Accept" recommendation.

Reviewer #2: All comments have been addressed

2. Is the manuscript technically sound, and do the data support the conclusions?

Reviewer #2: Yes

3. Has the statistical analysis been performed appropriately and rigorously? 

Reviewer #2: Yes

4. Have the authors made all data underlying the findings in their manuscript fully available?

Reviewer #2: Yes

5. Is the manuscript presented in an intelligible fashion and written in standard English?

Reviewer #2: Yes

6. Review Comments to the Author

Reviewer #2: Authors have explicitly addressed the identified issues regarding privacy. While re-identification risk have not been completely removed, authors have put in place necessary safeguards, including local LLM deployment. Authors have included notes about potential improvements, allowing the research community to build on their basis. Everything together, included acknoweledged limitations, this paper may be considered ready for publication.

7. PLOS authors have the option to publish the peer review history of their article (what does this mean?). If published, this will include your full peer review and any attached files.). If published, this will include your full peer review and any attached files.

.

Reviewer #2: No

---

## [Editor Report · Acceptance letter]

PONE-D-25-60609R3

PLOS One

Dear Dr. Dharma,

I'm pleased to inform you that your manuscript has been deemed suitable for publication in PLOS One. Congratulations! Your manuscript is now being handed over to our production team.

Kind regards,

on behalf of

Dr. Issa Atoum

Academic Editor

PLOS One